# ALIGNEDEDIT: PROMPT-ALIGNED WEAK GUIDANCE FOR TEXT-GUIDED IMAGE EDITING

## ABSTRACT

Text-guided image editing has advanced rapidly, yet most approaches still rely on classifier-free guidance. We introduce ALIGNEDEDIT for semantic image editing, employing semantically weak guidance to produce natural edits that align with the instruction prompt. CFG is a de facto standard guidance technique that use an uncondition model to steer sampling toward the positive condition and amplify its signal. However, this mechanism that use condition and uncondition model that misalend in semantic space induce over-editing, artifacts, and unintended changes. ALIGNEDEDIT employs aligned yet semantically weak guidance, preventing error accumulation and producing faithful edits without unintended modifications, resulting in a more natural appearance. To obtain an aligned yet semantically weak model, ALIGNEDEDIT identifies semantically strong tokens in each attention block and attenuates their embeddings to reduce semantic strength. Because the semantically weak model is derived directly from the model itself, no explicit negative prompt is required, making the method substantially less sensitive to prompt choice. We apply our guidance to two diffusion-based editing models, CosXL and Kontext. Across diverse benchmarks of Emu-Edit for real-image editing, HQ-Edit for synthetic editing, and ImgEdit-Bench for multi-turn editing, our method yields edits that are more natural and more faithfully aligned with the prompt.

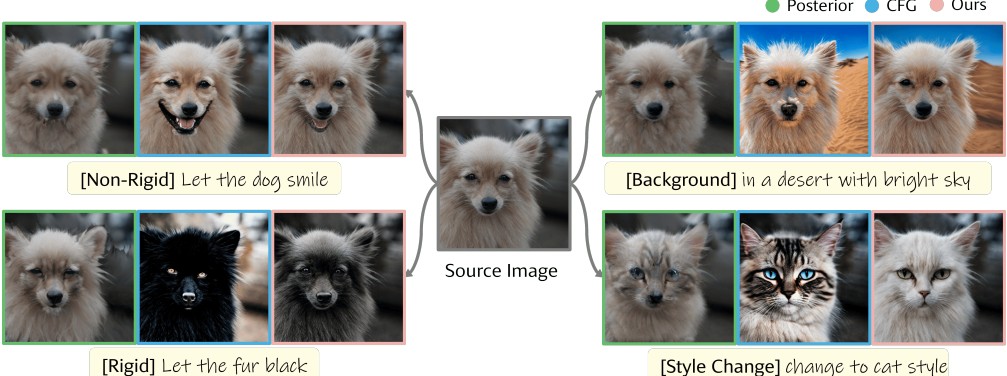

Figure 1: Visual comparison of three sampling methods: Posterior, CFG, and SWG. Posterior sampling often fails to follow the editing prompt and yields low-quality results. CFG causes over-editing, such as exaggerated smiles, unintended structural changes to the source image, and other unnatural artifacts. In contrast, our ALIGNEDEDIT produces edits that are faithful to the prompt and visually natural, preserving the original structure even under background or color changes.

## 1 INTRODUCTION

Text-instruction–based image editing (TIE) has advanced rapidly, enabling precise manipulation of objects, attributes, backgrounds, and even entire scenes via natural language (Brooks et al., 2023; Wei et al., 2025; Labs et al., 2025; Pathiraja et al., 2025). These approaches are practical because they avoid explicit masks (Couairon et al., 2022;

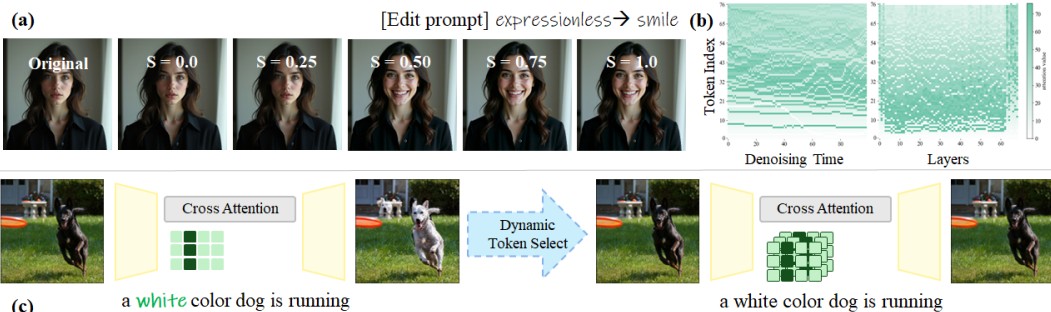

Figure 2: Semantic weak editing analysis. (a) With the text-editing prompt *let the woman smile* and scaling factor $s$, the effect is negligible at small. As increase $s$, the semantic smile appears. (b) At the same denoising timestep, the saliency ranking of tokens varies across different layers, while within a single layer, the attention distribution changes as timesteps evolve. (c) Zeroing out only the editing key-component token index is insufficient to nullify the corresponding semantic. Instead, at each inference step and for each layer, zeroing out salient tokens suppresses semantics effectively remove the target semantic.

Meng et al., 2022) and costly inversions (Huberman-Spiegelglas et al., 2024; Mokady et al., 2022), thereby reducing usage constraints and broadening applicability by leveraging the text prompt as the primary control signal. Nevertheless, despite this progress, current TIE development still leans heavily on scaling data and training (Parascandolo et al., 2013; Avrahami et al., 2022; Pathiraja et al., 2025; Geng et al., 2023a), and remains tied to conventional classifier-free guidance (CFG) (Ho & Salimans, 2022) at inference. In practice, repeated guidance steps exacerbate error accumulation (Ho & Salimans, 2022; Tian et al., 2025), and mode collapse (Chung et al., 2024) and the intrinsic misalignment between the conditional and unconditional branches (Hyung et al., 2024) often yields over-editing, artifacts, and unintended changes—ultimately harming both instruction fidelity and perceptual naturalness.

To address these drawbacks, several substitutes for CFG-style sampling guidance have been proposed. In image generation, AutoGuidance (Fu et al., 2024) trains auxiliary models and then steers the main model away from them, nudging samples toward higher-quality regions of the data manifold. PAG (Ahn et al., 2025) and SEG (Hong et al., 2023) achieve guidance by perturbing internal computations. SpatioTemporal Skip Guidance (Hyung et al., 2024) instead identifies an internal weak pathway by selectively skipping computations during sampling. In image editing, PostEdit (Parmar et al., 2023) has been introduced, incorporating a posterior scheme to govern diffusion sampling as an alternative to CFG. However, it still struggles to achieve precise, localized edits, and a definitive remedy remains elusive. Consequently, an optimal guidance method that can replace CFG for image editing has yet to be established. The primary driver of edits in TIE is the text prompt. Unlike pure image generation, our focus is semantic editing—prioritizing higher text fidelity while maintaining natural, coherent edits. Building on prior work, we steer the sampler using a low or suppressed-semantics model to improve instruction adherence without sacrificing realism.

In diffusion-based TIE, edits driven by the text prompt are mediated primarily through transformer cross-attention. Previous weak-model rely on negative or null prompts, but mismatch between the positive and negative branches can induce misalignment and deviate from the user intent. We instead construct a semantic weak model inside the network by modestly adjusting cross-attention. Empirically, we found out that using the same positive prompt and downscaling text-embedding before attention can yield an internally derived semantic weak model without retraining. However, naïvely zeroing all text embeddings suppresses semantics but degrades image quality. Rather attenuate all embedding, attenuating only the high-saliency tokens—identified from attention—reduces semantic strength while better preserving visual quality, thereby improving alignment without the drawbacks of external negative or null prompts.

We propose ALIGNEDEDIT, which construct semantically weak guidance (SWG) by adaptively identifying semantically strong token indices at each transformer layer and attenuating them during editing. We show that dynamically adaptive attenuating can introduce semantic suppresssion or weakening. As CFG use negative model as a weak model, ALIGNEDEDIT uses a model-internal, condition-aligned weak pathway to amplify the intended semantics without relying on extra negative prompts, improving prompt fidelity, preserving structure and induce more natural edit. By precomputing attention maps, we identify dominant tokens at each step and adaptively attenuate their contribution, which instantiates an internal semantically weak guidance model. We evaluate ALIGNEDEDIT and compare to prior work on EmuEdit (Sheynin et al., 2023), HQ-Edit (Hui et al., 2024), and ImgEdit-Bench (Ye et al., 2025), and obtained consistent gains in instruction fidelity, source-structure preservation, and perceptual naturalness.

## 2 RELATED WORK

**Text Instruction based Semantic Image Editing**  Semantic image editing aims to modify images according to a text instruction, ranging from subtle attribute changes to large-scale scene modifications (Sun et al., 2024). For prior work has explored mask-guided Prompt-to-Prompt (Hertz et al., 2022), BlendedDiffusion (Avrahami et al., 2022),Fisedit (Yu et al., 2024) or inversion-based (Mokady et al., 2022) or drag-based DragDiffusion (Shi et al., 2024), DragGAN (Pan et al., 2023), and concept-specific steering Concept Sliders (Gandikota et al., 2023), Prompt Sliders (Sridhar & Vasconcelos, 2024), TextSliders (Guerrero-Viu et al., 2024), AdaptiveSlider (Jain et al., 2025). Our focus is text instruction only editing that does not use any masks or inversion.

In TIE, earlier approaches relied on text-conditioned GANs SF-GAN (Yang et al., 2025), EditGAN (Ling et al., 2021) or CLIP-based (Patashnik et al., 2021), but these methods suffered from low generalization (Sun et al., 2024). In TIE, early work relied on text-conditioned GANs—SF-GAN (Yang et al., 2025) and EditGAN (Ling et al., 2021)—or CLIP-based methods such as StyleCLIP (Patashnik et al., 2021), but these approaches showed limited generalization (Sun et al., 2024). More recently, diffusion-based methods—including InstructPix2Pix (Brooks et al., 2023), InstructDiffusion (Geng et al., 2023b), RefEdit (Pathiraja et al., 2025), UltraEdit (Zhao et al., 2024), CosXL (Wei et al., 2025), and Kontext (Labs et al., 2025)—together with training-free inference schemes (e.g., tailored sampling schedules (Wang et al., 2025) and attention manipulation (Avrahami et al., 2025)) have delivered markedly stronger performance and broader applicability.

**Guidance Sampling**  CFG (Ho & Salimans, 2022) use null text prompt to improve sample quality. However, disaligned between condition and uncondition model cause reduced sample diversity (Chung et al., 2024; Karras et al., 2024), increased sampling trajectory curvature (Chung et al., 2024), error accumulation (Tian et al., 2025) resulting in skewed or oversaturated, and unnatural images. Training based AutoGuidance (Fu et al., 2024), and training-free based of PAG (Ahn et al., 2025) and SAG (Hong et al., 2023), and CFG++ (Chung et al., 2024) is several approaches that replace the unconditional model.

Adaptive Scaling (Malarz et al., 2025), Gradient-Free Classifier (Shenoy et al., 2024) extend CFG to mitigate its drawbacks by proposing more adaptive sampling schemes. Beyond images, spatiotemporal skip guidance has been proposed as a training-free replacement for CFG in video diffusion (Hyung et al., 2024), and PostEdit  (Tian et al., 2025) improve zero-shot editing fidelity while preserving structure.

## 3 ALIGNEDEDIT METHOD

### 3.1 CLASSIFIER-FREE GUIDANCE AND EDITING FORMULATION

In CFT, during training, the denoiser $\epsilon_\theta$ learns both conditional and unconditional settings by randomly replacing the text $c_T$ with a null token $\varnothing$. At inference, guidance is applied as

$$\hat{\epsilon}_\lambda(x_t, c_T) = \epsilon_\theta(x_t, c_T) + \lambda\big(\epsilon_\theta(x_t, c_T) - \epsilon_\theta(x_t, \varnothing)\big), \tag{1}$$

where $\lambda$ controls the guidance strength.

We are focusing on image editing. In text-guided editing, conditioning is extended to both the text prompt and the input image, with either randomly replaced by a null input during training to support both conditional and unconditional settings. This formulation yields three score estimates within a single forward pass:

$$\hat{\epsilon}(x_t) = \epsilon_\varnothing + \lambda_{\text{text}}(\epsilon_{\text{text}} - \epsilon_{\text{image}}) + \lambda_{\text{image}}(\epsilon_{\text{image}} - \epsilon_\varnothing), \tag{2}$$

where $\lambda_{\text{text}}$ and $\lambda_{\text{image}}$ control the guidance strength for text and image conditions respectively. Under standard CFG, the guidance direction is obtained by contrasting the unconditional prediction computed with null image and null text. Instead, we contrast it with a semantic weak model of the same instruction, the semantic weak guidance (SWG).

Our SWG for editing becomes

$$\begin{aligned}
\hat{\epsilon}_\theta(z_t, z_I, z_c; s_I, s_T) = {} & \epsilon_\theta(z_t, \varnothing_I, z_c^{\text{weak}}) \\
& + \lambda_{\text{text}}\left[\epsilon_\theta(z_t, z_I, z_c) - \epsilon_\theta(z_t, z_I, z_c^{\text{weak}})\right] \\
& + \lambda_{\text{image}}\left[\epsilon_\theta(z_t, z_I, z_c^{\text{weak}}) - \epsilon_\theta(z_t, \varnothing_I, z_c^{\text{weak}})\right].
\end{aligned} \tag{3}$$

where $\varnothing_I$ denote null image.

### 3.2 Designing of Semantic Weak Model

To design semantic weak model, we conduct a scaling experiment where text embeddings in each transformer block are multiplied by a scalar $s$ before cross-attention. By sweeping $s \in \{0, 0.25, 0.5, 0.75, 1.0\}$, we evaluate the effect of semantic scaling using CLIP directional similarity ($\text{CLIP}_{\text{dir}}$) (Gal et al., 2021) and ImageReward (Xu et al., 2023). We use 200 images generated with Flux, and for editing we pair each image with 10 GPT-generated prompts (OpenAI, 2024). The prompts span four categories—non-rigid, rigid, background, and add/remove—yielding 2,000 edit instructions in total.

As shown in Figure 2 (a), setting $s = 0$ suppresses prompt semantics with only a slight drop in image quality while $0 < s < 1$ progressively attenuates semantics up to a threshold. In practice, smaller $s$ weakens or suppresses edits and larger $s$ amplifies them, providing an intuitive controller for semantic strength. This experiment show that a semantically weak model can be obtained internally. As too small scaling parameter degrades image quality, the scale should be chosen optimal to suppress or weaken semantics only as much as necessary, balancing prompt fidelity and visual quality. Here, our question is **how can we suppress semantic strength while minimizing any degradation in perceptual quality?**

### 3.3 Dynamic of token saliency

Contrary to a naïve assumption, the target token is often not the most salient, and semantic strength varies markedly across layers, timesteps, and tokens. Token influence during editing is neither static nor uniform.

In Figure 2 (c) left show an edit with the prompt *a white color dog is running* and zeroing the token embedding corresponding to *white* (token index $j$=2). Despite ablating this token, the edited output remains *white*, indicating that the semantics of *white* are distributed across multiple tokens rather than concentrated in a single index.

Figure 2(b) presents the token-saliency map, obtained by computing attention and, for the corresponding token, taking the average over the same heads. The left of (b) shows token saliency over denoising time within the same layer, while the right shows saliency across different blocks at the same inference time.

These observations suggest that semantic strength varies dynamically across denoising time and layers. We find that efficiently weakening semantics requires dynamically selecting the

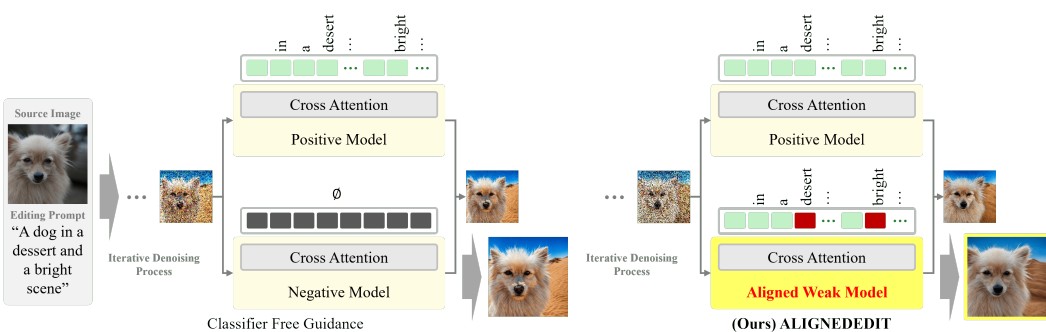

Figure 3: ALIGNEDEDIT overview. Left: conventional CFG. Right: an ALIGNEDEDIT transformer block, where the most salient tokens are selectively attenuated before attention. The visualization illustrates a single layer as an example.

token indices with high saliency. Details of the experiment is on Appendix D. Guided by these findings, our method builds a semantic weak model by adaptively attenuating the most dominant, that is semantically strong tokens at each transformer block and timestep.

## 3.4 ALIGNEDEDIT PROCESS

Given an source image $x$ and an editing prompt $c$, we aim to generate a modified image $\hat{x}$ that preserves the structure of $x$ while aligning with $c$. Especially, using DiT based model, pure gaussian noise $y$ is denoised to $\hat{x}$ through iterative denoising process. Different from CFG guidance, which uses a unconditional model as a null prompt unrelated to $c$, we leverage the same prompt $c$ to construct a semantic weak model. Although the semantic weak model and the strong model use the same prompt $c$, the semantic weak model reduces the semantics of $c$ via saliency-based token attenuation.

**Positive Logit Prediction** The input image $x$ and text $c$ are encoded to $z_I$ and $z_c$, respectively. We concatenate $y$ with $z_I$, i.e., $\text{concat}(y, z_I)$, and feed the result into the model, while $z_c$ interacts with the visual tokens inside the transformer. Using the same diffusion process, after a single denoising step we compute the positive logit from the resulting representation.

**Semantic Weak Logit Prediction** Analogous to the positive branch, the semantic weak logits are computed using the same instruction $c$, but with a weakened textual embedding. Inside each transformer block, cross-attention is applied between the visual tokens $\text{concat}(y, z_i)$ and the text tokens. The cross-attention scores are

$$A^{(h)} = \text{softmax}\left(\frac{Q^{(h)}K^{(h)\top}}{\sqrt{d}}\right) \in \mathbb{R}^{L_v \times L_c}, \tag{4}$$

where $Q^{(h)} \in \mathbb{R}^{L_v \times d}$ and $K^{(h)} \in \mathbb{R}^{L_c \times d}$ are the query and key matrices for head $h$, and $L_v$ and $L_c$ is the number of pixels and the text tokens respectively. Then per-token saliency $s_j$ for each text token $j \in \{1, \ldots, L_c\}$ is obtained by averaging attention over all visual positions and heads:

$$s_j = \frac{1}{H\,L_v} \sum_{h=1}^{H} \sum_{i=1}^{L_v} A_{i,j}^{(h)}. \tag{5}$$

The embeddings of the selected top-$K$ tokens are scaled down, and the transformer operations in this block are recomputed. Here important is that semantic weak logit and positive logits is based on same text prompt. The only difference is that in semantic weak logit prediction, only text embeddings of high saliency tokens are only attenuated. Finally, the

semantic weak attention output is obtained by applying the weights to the values. The block output is finalized via the residual connection, completing the transformer-block computation. The same procedure is applied across all blocks of the semantic weak model for $\epsilon_\theta(z_t, z_I, z_c^{\text{weak}})$ and $\epsilon_\theta(z_t, \varnothing_I, z_c^{\text{weak}})$ in Eq. 3.

**Application of CFG Formula for Editing** Following Eq. 3, the diffusion score at each step is expressed by combining the condition score, the image-conditioned with semantic, and the semantic weak conditioned score. This stepwise guidance is directly applued into during every inference step. The complete ALIGNEDEDIT algorithm is provided in Appendix D.4.

## 4 EXPERIMENTS

### 4.1 OVERVIEW

**Models** We evaluate our ALIGNEDEDIT on two CFG based TIE models, CosXL (Wei et al., 2025) and Kontext (Labs et al., 2025). CosXL is built on SDXL (Podell et al., 2023) and comprises 140 transformer blocks (70 self-attention and 70 cross-attention). We also evaluate the FLUX based (Yang et al., 2024) TIE model, Kontext, which employs a two-stream architecture with 57 transformer blocks—19 single-stream and 38 multi-stream—and. As Kontext, unlike SDXL, performs both self- and cross-attention within every transformer block, we apply SWG to all 57 blocks.

**Comparison models** We compare three sampling strategies: CFG, PostEdit (Tian et al., 2025), and our proposed SWG. For CFG guidance, we first consider StableFlow (Avrahami et al., 2025), which is built upon the state-of-the-art FLUX image generation model and performs editing through attention control. We also evaluate IceEdit (Zhang et al., 2025), which is also based on FLUX but leverages LoRA training to adapt the image generation framework for editing with the initial noise control. Second the PostEdit incorporate posterior sampling to enhance reconstruction for image editing.

**Datasets** We evaluate our models on three image-editing benchmarks. First, we use the EmuEdit (Sheynin et al., 2023) test set, which contains 3,590 examples spanning seven categories. Second, we include HQ-Edit (Hui et al., 2024), a large-scale, high-resolution instruction-based synthetic dataset (about 200,000 edits) constructed via a scalable pipeline leveraging GPT-4V and DALL·E 3 (OpenAI, 2023). Unlike EmuEdit, HQ-Edit is fully synthetic, enabling fair and extensive comparison of model performance under large-scale synthetic conditions. Third, we adopt ImgEdit-Bench (Ye et al., 2025), which—unlike the previous two single-turn benchmarks—supports multi-turn editing with three task types: content memory, content understanding, and version backtracking.

### 4.2 EXPERIMENTS ON EMU EDIT

On Emu Edit, following the protocol of Emu Edit, we evaluate edit results along three criterias: instruction fidelity, source preservation, and naturalism. For instruction fidelity, we use $\text{CLIP}_{\text{dir}}$ to assess the similarity between caption changes and image changes, and CLIP output similarity ($\text{CLIP}_{\text{out}}$) for measuring how well the edited image matches the output caption. For source preservation, we use CLIP image similarity ($\text{CLIP}_{\text{img}}$) between the edited and input images and SSIM to quantify how well non-editing target content is preserved. For naturalism, we report ImageReward (ImgRWD), Inception Score (IS) (Salimans et al., 2016), and Fréchet Inception Distance (FID).

Table 1 and Figure 4 present the results. StableFlow exhibits low $\text{CLIP}_{\text{dir}}$ despite high SSIM and low FID, with frequent editing failures—evidence that attention-based editing is not universally effective across diverse semantic edits. IceEdit attains relatively high $\text{CLIP}_{\text{dir}}$, indicating strong semantic performance, yet still fails to fully follow certain instructions (e.g., *teddy bear*). PostEdit also achieves moderately higher $\text{CLIP}_{\text{dir}}$, but its ImageReward and FID suggest that, while reconstruction improves, the sampling strategy remains ill-suited

| | Emu Edit Bench | | | | | | | Base model | Sampling | | |
|---|---|---|---|---|---|---|---|---|---|---|---|
| Model | CLIP$_{dir}$ ↑ | CLIP$_{out}$ ↑ | CLIP$_{img}$ ↑ | SSIM ↑ | ImgRWD ↑ | FID ↓ | IS ↑ | | CFG | SWG | Post |
| StableFlow | 0.021 | 0.211 | 0.978 | **0.982** | 0.367 | **170.099** | 5.918 | Flux | ✓ | | |
| IceEdit | 0.098 | 0.222 | 0.880 | 0.781 | 0.531 | 185.808 | 6.623 | Flux | ✓ | | |
| PostEdit | 0.054 | 0.240 | 0.821 | 0.641 | 0.140 | 274.799 | 8.982 | LCM | | | ✓ |
| CosXL | 0.086 | 0.222 | 0.853 | 0.753 | 0.525 | 214.728 | **10.790** | SDXL | ✓ | | |
| Kontext | 0.117 | 0.233 | 0.863 | 0.741 | 0.813 | 194.558 | 2.823 | Flux | ✓ | | |
| CosXL w/ SWG | 0.052 | **0.418** | **0.981** | 0.980 | 0.541 | 225.938 | 6.792 | SDXL | | ✓ | |
| Kontext w/ SWG | **0.429** | 0.337 | 0.967 | 0.845 | **0.997** | 190.347 | 3.785 | Flux | | ✓ | |
| | HQ-Edit Bench | | | | | | | ImgEdit-Bench | | | |
| Method | CLIP$_{dir}$ ↑ | CLIP$_{out}$ ↑ | CLIP$_{img}$ ↑ | SSIM ↑ | ImgRWD ↑ | FID ↓ | IS ↑ | (T1) Fidel ↑ | (T1) Nat ↑ | (T2) Fidel ↑ | (T2) Nat ↑ |
| StableFlow | 0.098 | 0.242 | **0.962** | **0.965** | 0.018 | **17.530** | 11.912 | 2.301 | 2.539 | 2.467 | 2.369 |
| IceEdit | 0.259 | 0.268 | 0.919 | 0.622 | 0.856 | 69.350 | 11.966 | 3.814 | 3.836 | 3.965 | 3.933 |
| PostEdit | 0.230 | 0.277 | 0.897 | 0.360 | 0.657 | 88.126 | 11.665 | 1.052 | 2.014 | 1.004 | 1.872 |
| CosXL | 0.280 | 0.274 | 0.893 | 0.683 | 0.939 | 75.495 | 11.379 | 3.652 | 3.734 | 2.656 | 2.623 |
| Kontext | 0.305 | **0.282** | 0.869 | 0.398 | **1.349** | 84.589 | 12.026 | 4.338 | 4.011 | 3.915 | 3.869 |
| CosXL w/ SWG | 0.280 | 0.275 | 0.901 | 0.754 | 0.861 | 107.238 | 11.685 | 4.012 | 3.856 | 2.875 | 2.986 |
| Kontext w/ SWG | **0.343** | 0.264 | 0.872 | 0.462 | 0.998 | 269.144 | **12.062** | **4.396** | **4.152** | **4.022** | **4.205** |

Table 1: Results on Three Benchmarks. Top: Emu Edit Bench. Top right: categorization of models by base architecture and sampling method. Bottom left: HQ-Edit Bench metrics. Bottom right: ImgEdit-Bench multi-turn performance, where T1 and T2 indicate Turn 1 and Turn 2, respectively, Fidel and Nat means Fidelity and Naturalism. The best scores in each category are highlighted in bold.

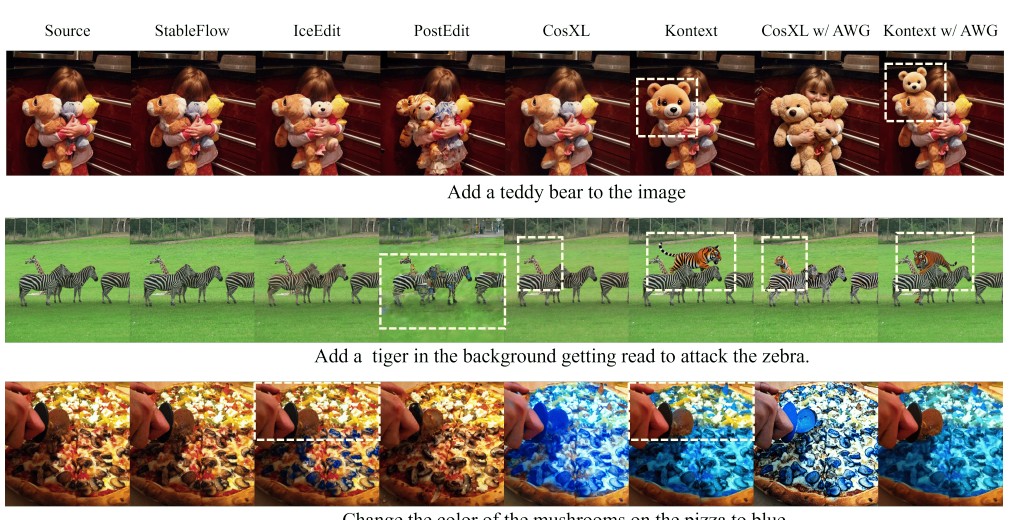

Figure 4: Comparison with baseline models on the Emu Edit test set. Our method shows superior performance over the baselines in both instruction-following accuracy and preservation of non-edited regions.

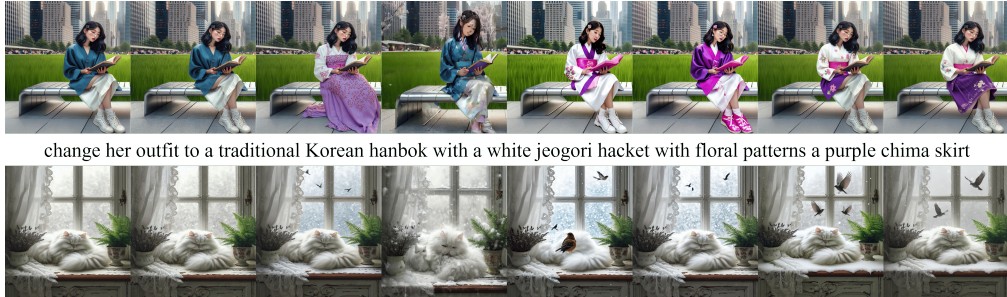

Figure 5: Comparison with baseline models on the HQ-Edit test set. The leftmost column is the source, and the remaining columns follow the same model order as in Figure 4.

for editing. By contrast, ALIGNEDEDIT faithfully realizes the *teddy bear* instruction, delivering higher instruction fidelity while preserving non-edited content.

In the second example, ALIGNEDEDIT produces a more natural edit: the tiger generated by the Kontext baseline appears awkward, while ALIGNEDEDIT conveys a more convincing sense of attack. In the third example, ALIGNEDEDIT remains faithful to the instruction while constraining modifications to the relevant region. Specifically, CosXL shows color spill beyond the mushroom, whereas our sampling confines the color change strictly to the mushroom. IceEdit detects the mushroom but edits only one side of the pizza, resulting in incomplete editing. The Kontext baseline introduces an unnecessary yellow tint, which is mitigated in ALIGNEDEDIT.

**User Study** We conducted a qualitative preference study. For each prompt, participants are required to compare edits produced by ALIGNEDEDIT, CFG, and PostEdit and selected the preferred output along three criteria: instruction fidelity, source preservation, and naturalness. As shown in Figure 6, ALIGNEDEDIT was chosen frequently than both baselines. Further details of the protocol are provided in Appendix E.

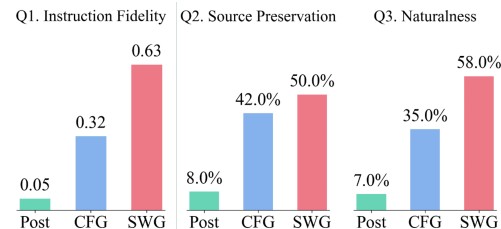

Figure 6: User study results. Participants preferred ALIGNEDEDIT over CFG and PostEdit.

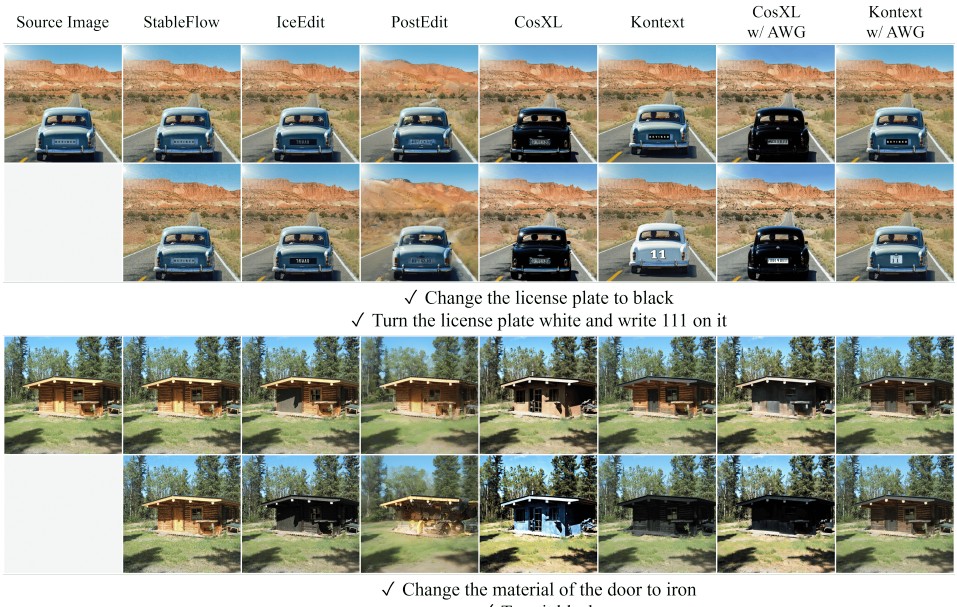

✓ Change the license plate to black
✓ Turn the license plate white and write 111 on it

✓ Change the material of the door to iron
✓ Turn it black

Figure 7: Edits on ImgEdit-Bench with ALIGNEDEDIT and baselines. In both examples, the first row shows Turn 1 and the second row shows Turn 2 and the Turn 2 input is the output of Turn 1.

### 4.3 EXPERIMENT ON HQ-EDIT BENCH

Evaluation results on the HQ-Edit benchmark are reported in Table 1 and Figure 5. Table 1 shows consistently higher performance on synthetic data than on real images. In Figure 5, StableFlow shows virtually no change. IceEdit alters the original structure, and PostEdit often collapses the image severely. With CosXL under CFG, facial identity drifts and the purple color is not reliably applied; in contrast, ALIGNEDEDIT preserves the face and accurately reflects the purple edit. The same advantage appears on Kontext. The skirt color is rendered as a clear purple, indicating strong adherence to the editing prompt. In the second example, the superiority of ALIGNEDEDIT is even clearer. With CosXL, the

cat degrades and the *a couple of birds* instruction is not realized, whereas ALIGNEDEDIT renders the birds correctly without collapsing the scene. On Kontext, baseline outputs often depict three birds regardless of the phrase *a couple of,* but ALIGNEDEDIT consistently produces exactly two, markedly improving instruction fidelity.

### 4.4 EXPERIMENT ON IMGEDIT BENCH

We evaluate multi-turn editing on ImgEdit-Bench. The benchmark pairs a global directive with turn-specific instructions. Because the evaluated models are designed for single-turn editing, we adapt the setting as follows: for Turn 2, we prepend the global directive to the turn-2 command, and we use the output of Turn 1 as the input to Turn 2. Further experimental details are provided in AppendixE.2. Table 1 shows the results. StableFlow and PostEdit attains low scores on both Turn 1 and Turn 2, reflecting limited editing ability and naturalness. IceEdit surpasses CosXL on both turns; however, CosXL combined with SWG yields sizable gains, reaching results comparable to IceEdit. In Figure 7 first example of license editing, CFG sampling with CosXL and Kontext fails to follow the instruction, whereas ALIGNEDEDIT on CosXL renders a white plate correctly labeled *111,* and on Kontext yields a much more coherent edit.

**Ablation Study**  We conduct ablation experiments to analyze two key factors in constructing the proposed weak model, the attenuation scalar applied to token embeddings, and the number of tokens subject to attenuation. If the scaling factor equals 0 and the number of tokens to full text embedding length, method convert to CFG. Figures 8 and 9 summarize the result. In Figure 8 (a), performance generally improves as we apply mild-to-moderate attenuation to salient tokens. However, all token zeroing out removes useful signal and degrades results. This trend is visible in Figure 9 with a small attenuation (e.g., 0.2) the word *Tray* is rendered more reliably than under full zero-out, as seen in the second column. In Figure 8 (b), increasing the number of attenuating tokens initially helps, but beyond a threshold it begins to hurt. Over-attenuating too many tokens lowers saliency globally—including for edit-relevant tokens—so the effect is not a mere relative dilution but a uniform suppression, rendering it unsuitable for semantic weak guidance. Therefore, selectively attenuating only the most edit-relevant tokens is more effective than blanket attenuation for constructing a semantic weak model.

**Conclusion**  We present ALIGNEDEDIT, a sampling strategy for TIE. Conventional, CFG guidance is a conventional sampling in TIE, which cause over or unnatural editing. Unlike CFG that unconditional branch can introduce spurious semantics during editing, ALIGNEDEDIT suppresses unintended transfer

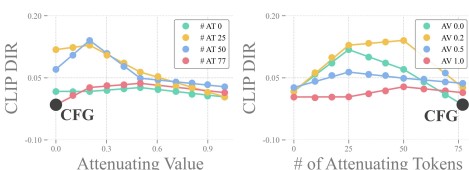

Figure 8: Clip directional score across attenuating value and the number of attenuating tokens. AT denotes the number of attenuated tokens, and AV is the attenuation value.

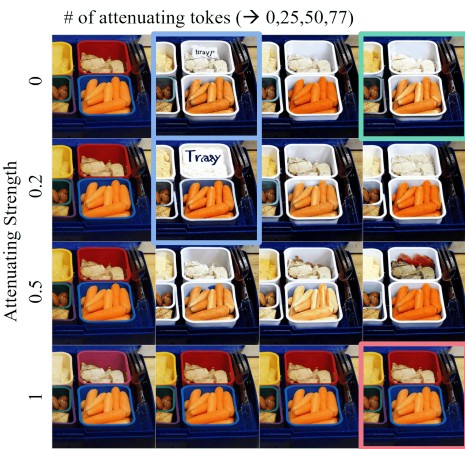

Figure 9: Qualitative effects across $A_V$ and $A_T$. A green border denotes CFG, a pink border denotes no guidance, and a cyan border denotes a successful edit.

by attenuating the currently dominant text tokens, identified via cross-attention saliency, without any retraining. We motivate this design with a theoretical analysis of CFG and through experiments showing more faithful and natural edits. Across EmuEdit, HQ-Edit, and ImgEdit-Bench, ALIGNEDEDIT consistently outperforms CFG and delivers strong results on CosXL and Kontext, positioning it as a practical, general replacement for CFG in image editing.

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
