# ALIGNEDEDIT: Prompt-Aligned Weak Guidance for Text-Guided Image Editing
## Supplementary Material

This appendix provides a clearer explanation of the CFG (Ho & Salimans, 2022) foundation of our experiments, the design of the Semantic Weak Model along with its theoretical analysis, and the experimental details and qualifications.

- A. Limitations.

- B. Future Research.

- C. CFG in Image Editing.

- D. Experiments on Semantic Weak Model and Theoretical Analysis

- E. Experiments Details and Qualitative Results

## A  LIMITATIONS

Since our method requires manual selection of the attenuation strength $s$ and the number of tokens (Top-$K$), further advancements in automatic selection could enhance the ALIGNEDEDIT algorithm. Just as the guidance scale in CFG affects edited images, Top-$K$ and the attenuation scale also influence results, so further investigation of these hyperparameters could offer deeper understanding and lead to further improvements.

## B  FUTURE RESEARCH

While many studies diagnose the limitations of CFG and propose remedies, most advances focus on image generation (Fu et al., 2024; Ahn et al., 2025; Hong et al., 2023), leaving image editing comparatively underexplored. Posterior-based approaches (Tian et al., 2025) point in a promising direction, but our experiments show that they remain insufficient for robust editing. We propose ALIGNEDEDIT as an alternative to CFG, tailored for the editing setting. Since image editing inherently involves not only the text prompt condition but also a source image condition, deeper research on aligned negative image conditions could further improve performance. Although our current framework focuses on text guidance, extending the approach to leverage image-based conditions represents an important next step.

## C  OPTIMAL CLASSIFIER-FREE GUIDANCE

### C.1  CLASSIFIER GUIDANCE

Classifier Guidance (Dhariwal & Nichol, 2021) augments the conditional score with the gradient of an auxiliary classifier. Let $\epsilon_\theta(z_\lambda, c) \approx -\sigma_\lambda \nabla_{z_\lambda} \log p(z_\lambda \mid c)$ be the conditional score and $p_\theta(c \mid z_\lambda)$ a classifier trained at noise level $\lambda$. CG modifies the score as

$$\tilde{\epsilon}_\theta(z_\lambda, c) = \epsilon_\theta(z_\lambda, c) - w\, \sigma_\lambda\, \nabla_{z_\lambda} \log p_\theta(c \mid z_\lambda),$$

which corresponds to sampling approximately from

$$\tilde{p}_\theta(z_\lambda \mid c) \;\propto\; p_\theta(z_\lambda \mid c)\, \big[p_\theta(c \mid z_\lambda)\big]^w.$$

Intuitively, CG increases the weight of regions where the classifier assigns high likelihood to the target class.

## C.2 CLASSIFIER-FREE GUIDANCE

As CG relies on an external classifier, CFG was introduced as a remedy. CFG achieves effects similar to CG without relying on any external classifier. A single diffusion network is trained to produce both unconditional and conditional scores by randomly dropping the condition during training (i.e., feeding the null token $\varnothing$). At sampling time, the guided score is formed by a linear combination

$$\tilde{\epsilon}_\theta(z_\lambda, c) = (1 + w)\,\epsilon_\theta(z_\lambda, c) - w\,\epsilon_\theta(z_\lambda),$$

where $w \geq 0$ controls the strength of guidance. If $\epsilon_\theta$ perfectly matched the true conditional and unconditional scores, e.g.,

$$\epsilon^*(z_\lambda, c) \propto -\nabla_{z_\lambda} \log p(z_\lambda \mid c) \quad \text{and} \quad \epsilon^*(z_\lambda) \propto -\nabla_{z_\lambda} \log p(z_\lambda),$$

then their difference would satisfy

$$\epsilon^*(z_\lambda, c) - \epsilon^*(z_\lambda) \propto -\nabla_{z_\lambda} \log \frac{p(z_\lambda \mid c)}{p(z_\lambda)} \propto -\nabla_{z_\lambda} \log p_i(c \mid z_\lambda),$$

where $p_i(c \mid z_\lambda) \propto p(z_\lambda \mid c)/p(z_\lambda)$ can be viewed as an implicit classifier. This observation motivates the CFG form above: by amplifying the conditional score and subtracting the unconditional one, the update approximately follows the gradient that increases the implicit classifier likelihood of the target condition. In practice, tuning $w$ trades diversity for fidelity and often improves standard generation metrics (e.g., FID and IS) compared to unguided sampling, while avoiding the expense of training noise-level–specific external classifiers.

## C.3 CFG IN IMAGE EDITING

In image editing, unlike pure image generation, the model must respond to two signals: the input image $c_I$ and the text instruction $c_T$. Following InstructPix2Pix (Brooks et al., 2023), we extend CFG by pairing the text null token with an image-null condition, and train a single shared network to handle three branches:

$\epsilon_\theta(z_t, \varnothing, \varnothing)$ (unconditional), $\epsilon_\theta(z_t, c_I, \varnothing)$ (image only), and $\epsilon_\theta(z_t, c_I, c_T)$ (image+text). At sampling time, these are combined as

$$\tilde{\epsilon}_\theta(z_t, c_I, c_T) = \epsilon_\theta(z_t, \varnothing, \varnothing) + s_I\big[\epsilon_\theta(z_t, c_I, \varnothing) - \epsilon_\theta(z_t, \varnothing, \varnothing)\big]$$
$$+ s_T\big[\epsilon_\theta(z_t, c_I, c_T) - \epsilon_\theta(z_t, c_I, \varnothing)\big], \tag{C1}$$

where $s_I, s_T \in \mathbb{R}$ control the relative influence of the image and text, respectively. Setting $s_I = s_T = 0$ yields unconditional sampling; choosing $s_I = 1$ and $s_T = 0$ reduces to image-conditioned denoising without editing; and setting $s_I = s_T = 1$ recovers the standard two-condition predictor used for conditioned editing.

For CFG to truly substitute CG in image editing, the residual score increments must be strictly proportion:

$$\Delta_I(z_t) = \epsilon_\theta(z_t, c_I, \varnothing) - \epsilon_\theta(z_t, \varnothing, \varnothing), \qquad \Delta_T(z_t) = \epsilon_\theta(z_t, c_I, c_T) - \epsilon_\theta(z_t, c_I, \varnothing),$$

with

$$\Delta_I(z_t) \propto \nabla_{z_t} \log p(c_I \mid z_t), \qquad \Delta_T(z_t) \propto \nabla_{z_t} \log p(c_T \mid c_I, z_t).$$

If this proportionality fails, the linear mixing in Eq. equation D2 becomes biased due to mis-alignment, and CFG cannot faithfully replace CG. In practice, to maintain high fidelity to the instruction prompt, it is required to increase the guidance weights $s_T$. However, higher guidance weights may actually induce misalignment—reducing naturalness and worsening FID and IS—thereby contravening the proportionality assumption (Tian et al., 2025). Conversely, too small weights lead to under edit, that the image degraded in instruction fidelity. Therefore, image editing requires an effective substitute for CG that avoids an external classifier, and ALIGNEDEDIT is motivated by precisely this need.

# D  Optimal Semantic Weak Model

An optimal negative model should steer updates toward the intended edit without relying on an external classifier. Unlike SpatioTemporal (Hyung et al., 2024) in video generation—which designs an inner model to preserve image quality, our editing objective is to construct an inner model that deliberately weakens semantics, thereby improving prompt fidelity.

From our experiments on semantic attenuation (Section 3.2), we observe that semantic weakness can be achieved directly from within the model through text embedding control. While Section 3.2 applies uniform control to all text embeddings, a more effective and efficient semantic weak model is obtained by selectively suppressing the edit-critical tokens that carry the core semantics.

In this section, we present our motivation experiments toward building semantic weak model and provide theoretical analysis of why this approach proves beneficial.

## D.1  Dynamic Attention Control

| Model | Score | Fixed | | | Dynamic | | |
|---|---|---|---|---|---|---|---|
| | | one token | condition tokens | padding tokens | Top K 1 | Top K #2 | Top K #3 |
| CosXL | $CLIP_{dir}$ | 0.311 | 0.170 | 0.240 | 0.056 | **0.043** | **0.035** |
| | ImgRWD | 1.753 | 1.746 | 1.749 | **-0.093** | **-0.093** | **-0.093** |
| | FID | 210.512 | 212.385 | 217.785 | 207.577 | 232.355 | 227.826 |
| | IS | 10.343 | 10.343 | 10.343 | 9.752 | 8.715 | 8.511 |
| Kontext | $CLIP_{dir}$ | 0.334 | 0.014 | 0.240 | 0.329 | **0.015** | **0.029** |
| | ImgRWD | 1.450 | 0.245 | 1.303 | 1.447 | **1.077** | **-0.264** |
| | FID | 113.578 | 125.852 | 123.723 | 111.511 | 185.452 | 137.682 |
| | IS | 12.343 | 12.343 | 13.343 | 11.752 | 10.340 | 10.121 |

Table D1: Zero-out strategies between fixed and dynamic. Sky-blue columns denote the two cases with the lowest $CLIP_{dir}$ scores and high FID

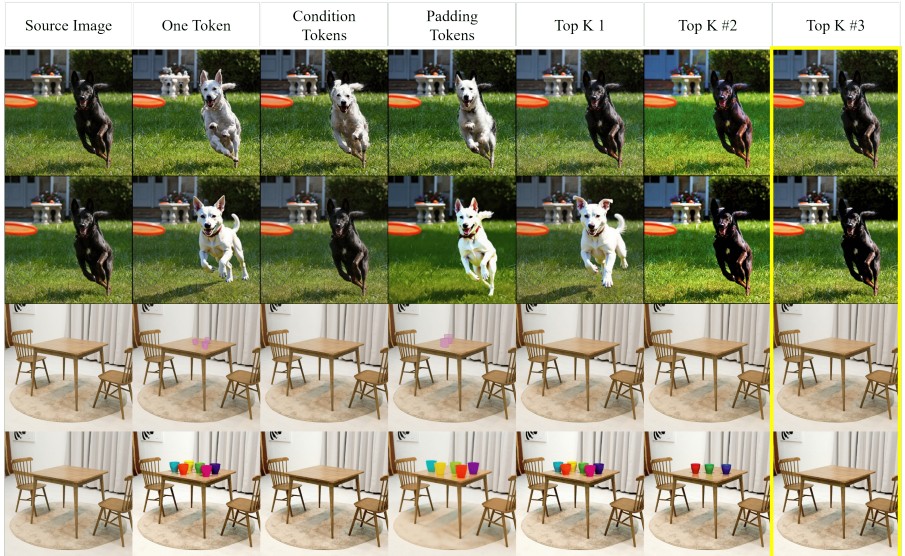

Figure D1: Visualization of edited image output of zero-out strategies. Cases highlighted in yellow bold indicate the case where the meaning of the editing prompt has completely disappeared.

A semantic weak model, serving as a substitute for the negative model, should align with the direction of the condition model while keeping semantic strength weakened or suppressed, and it should exhibit minimal degradation in quality. In instruction-guided editing, a few tokens typically carry the core semantics of the change. For example, when converting

a *black dog* to a *white dog*, the token *white* is the edit-critical cue. Whereas the source image encodes a black dog, while the text requests white. By selectively weakening such edit-critical tokens, we may expect the editing to become weaker. However, in practice, the semantic meaning of the white token is propagated through surrounding tokens (Li et al., 2024), so a more strategic approach is required to build an effective semantic weak model. To this end, we conduct a simple experiment.

We compare fixed and dynamic attenuation strategies on two diffusion editing bases, Kontext (Labs et al., 2025) and CosXL (Wei et al., 2025). For the fixed baselines, we (i) zero a manually selected key token $t^*$, (ii) zero all condition-text tokens together with `[CLS]`, and (iii) zero all padding tokens. In contrast, the dynamic strategy ranks tokens by attention saliency at each layer and zeros the top-$K$ tokens. We sweep $K \in 1, 10, 50$ for CosXL and $K \in 1, 100, 250$ for Kontext.

We evaluate prompt fidelity, structural preservation, and perceptual quality with CLIPdir, ImageReward (ImgRWD), FID, and IS. Table D1 show the result. In Table D1, the cases where CLIPdir and ImgRWD (Xu et al., 2023) drop most strongly high light. When only key tokens or condition tokens are zeroed (second row), CLIPdir remains high, indicating that the semantic of editing prompt remain. In contrast, zeroing high-saliency tokens (rows 6 and 7) causes CLIPdir to decrease significantly. Figure D1 is visualization of edited imaged in each case.

As can be seen, the fixed position based token zero out exhibits unreliable suppression of semantics. For CosXL, in the first example of transforming a *black dog* to *white dog*, the semantic meaning of *white* was not sufficiently weakened when zeroing condition tokens. However, under the dynamic saliency based zeroing strategy, removing even the single most salient token (top-1) caused the effect to vanish. Kontext demonstrated stronger semantic redundancy compare to CosXL. Even when the top-1 token was zeroed out, the meaning of *white* persist. With more saliency tokens to zero-out, *white* semantic vanished.

The results show that zeroing high-saliency tokens effectively reduces semantic strength. In our experiments, the semantic weak model uses the same prompt as the condition model; consequently, alignment between the condition model and the semantic weak model is substantially higher than in standard CFG, which relies on a negative or null prompt. In the next section, we analyze why this aligned negative condition can better preserve prompt fidelity and thereby reduce unintended edits.

## D.2 THEORETICAL ANALYSIS

In CFG, the text-conditioned branch $p_c(z) = P(z \mid c)$ is contrasted with an unconditioned null branch $p_{\text{null}}(z) = P(z \mid \varnothing)$, and the score upade

$$
\begin{aligned}
\tilde{\epsilon}_\theta^*(z, c) &= \epsilon_\theta(z, \varnothing) \\
&\quad + s\big[\epsilon_\theta(z, c) - \epsilon_\theta(z, \varnothing)\big] \\
&= (1 - s)\,\epsilon_\theta(z, \varnothing) + s\,\epsilon_\theta(z, c) \quad \text{(D1)}
\end{aligned}
$$

is biased toward the unconditional direction $\epsilon_\theta(z, \varnothing)$. Consequently, the trajectory under $\tilde{\epsilon}_\theta^*$ is displaced from the mode favored by the conditional field $\epsilon_\theta(z, c)$, as illustrated by the green curve in the left panel of Figure D2. Intuitively, adding a misaligned reference pulls probability mass toward the null branch, so the effective peak moves away from the intended instruction.

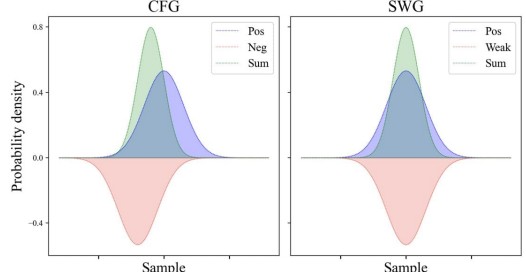

Figure D2: Comparison between CFG and SWG. In CFG, the unconditioned branch is misaligned with the positive branch, so guidance expands the distribution in a shifted direction. In SWG, a semantic weak branch is anchored to the same instruction, so guidance is centric around the condition mode.

The consequences of this displacement differ between generation and editing. In pure generation, shifting the peak causes samples to cluster near a new high-density region, which can improve fidelity and yield sharper images, although diversity may be slightly reduced.

Such a shift is often acceptable, or even beneficial, for generative tasks. In image editing, however, the situation is more delicate. The source image serves as an anchor that defines the starting distribution, and editing requires a precise transition toward the target distribution. If the conditional distribution itself is shifted, as in CFG, every denoising step introduces a small error (e.g., unintended color bleeding or geometric drift) (Tian et al., 2025). These errors accumulate across timesteps, ultimately leading to edited images that deviate from the intended instruction. To remove this misalignment, we replace the null model with a semantic weak model. Let

$$p_{\text{weak}}(z) = P(z \mid c^{\text{weak}})$$

denote this weak branch. The corresponding SWG update is then

$$\tilde{\epsilon}_\theta^{**}(z, c_{\text{edit}}) = \epsilon_\theta(z, c_{\text{weak}}) + s\left[\epsilon_\theta(z, c_{\text{edit}}) - \epsilon_\theta(z, c_{\text{weak}})\right] = (1-s)\,\epsilon_\theta(z, c_{\text{weak}}) + s\,\epsilon_\theta(z, c_{\text{edit}}).$$

In the case of image editing, the formulation becomes

$$\begin{aligned}
\tilde{\epsilon}_\theta(z_t, c_I, c_T) = \epsilon_\theta(z_t, \varnothing, c_{\text{weak}}) &+ s_I\left[\epsilon_\theta(z_t, c_I, c_{\text{weak}}) - \epsilon_\theta(z_t, \varnothing, c_{\text{weak}})\right] \\
&+ s_T\left[\epsilon_\theta(z_t, c_I, c_T) - \epsilon_\theta(z_t, c_I, c_{\text{weak}})\right],
\end{aligned} \tag{D2}$$

Since $p_{\text{weak}}$ and $p_c$ share the same semantic mode, guidance concentrates probability mass within that region rather than shifting across manifolds, sharpening the intended concept without displacement (right panel of Fig. D2, blue distribution to green distribution). Consequently, ALIGNEDEDIT amplifies the target attribute, improves prompt fidelity, localizes edits more precisely, and produces more natural results.

### D.3 COMPARISON WITH CFG IN EXPERIMENT

We compare CFG and SWG guidance using Emu Edit (Sheynin et al., 2023) benchmark which contain eight sub categories of adding objects, background change, color change, global change, local change, removal, style and text-related editing. The evaluation protocol is same as main experiment on Emu Edit. For CosXL, we use 50 sampling steps and a guidance scale of 5.0; for Kontext (Labs et al., 2025), we use 28 sampling steps and a guidance scale of 2.0.

Table D2 and Figure D3 is the result on CosXL.

While SWG improves prompt alignment overall, performance in the *Remove* category degrades.

This phenomenon stems from CosXL's pretraining. By applying SWG, alignment improves, yielding edits that better match the given prompt. Although most categories improve, the score for *Remove* decreases, likely because CosXL was not fully trained for removal; SWG further concentrates on objects explicitly mentioned in the prompt rather than suppressing them. These results yield two takeaways: (1) SWG delivers outputs more closely aligned with the given text, and (2) as alignment increases, the model's pretraining state becomes increasingly critical.

Figure D3 shows results on CosXL. Compared with CFG, SWG yields edits that are generally more natural or more accurate across most categories—for example, in *Text* the target glyph is better matched (highlighted with white dotted lines), and in *Background* the scene is rendered with fewer artifacts. However, performance degrades in the *Remove* category: stronger alignment makes the model more inclined to realize objects mentioned in the prompt, which can counteract removal. We expect this effect to diminish as the base model's removal capability improves.

Figure D4 presents results on Kontext. As with CosXL, SWG yields substantial gains on Kontext. Unlike CosXL, however, *Remove* edits are executed more cleanly, which implies more natural editing. Unlike CosXL, Kontext preserves the source less well, particularly in the *Background* and *Global* categories. By contrast, with SWG we observe that both editing fidelity and preservation are well satisfied.

| Model | Sampling | Add | Text | Back-ground | Color | Style | Global | Remove | Local |
|---|---|---|---|---|---|---|---|---|---|
| CosXL | CFG | 0.085 | **0.022** | 0.122 | 0.126 | 0.116 | 0.107 | 0.029 | 0.111 |
|  | SWG | 0.105 | 0.064 | 0.137 | 0.201 | 0.155 | 0.120 | 0.007 | 0.140 |
|  | Diff | **0.020** | 0.041 | **0.015** | **0.076** | **0.039** | **0.013** | -0.022 | **0.029** |
| Kontext | CFG | 0.136 | 0.086 | 0.136 | 0.187 | **0.096** | 0.110 | 0.085 | 0.102 |
|  | SWG | 0.204 | 0.097 | 0.139 | 0.344 | 0.226 | 0.198 | 0.158 | 0.105 |
|  | Diff | **0.068** | **0.011** | **0.003** | **0.158** | 0.131 | **0.087** | **0.073** | **0.003** |

Table D2: Category-wise $CLIP_{dir}$ comparison between CFG and SWG and their difference. Diff is SWG−CFG; blue shading marks SWG rows in columns 3–10. Bold indicates the larger improvement per category.

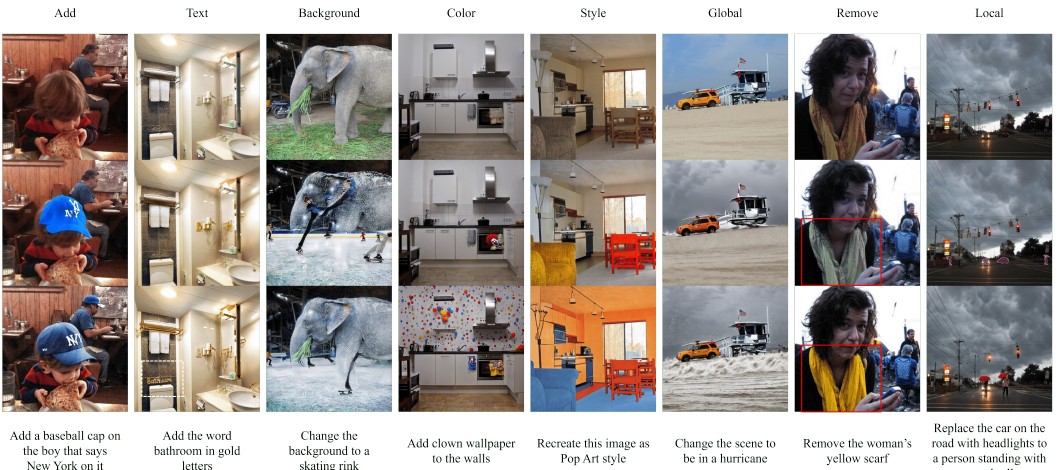

Figure D3: Visualization on CosXL comparing CFG and SWG. Rows from top to bottom: Source, CFG, and SWG.

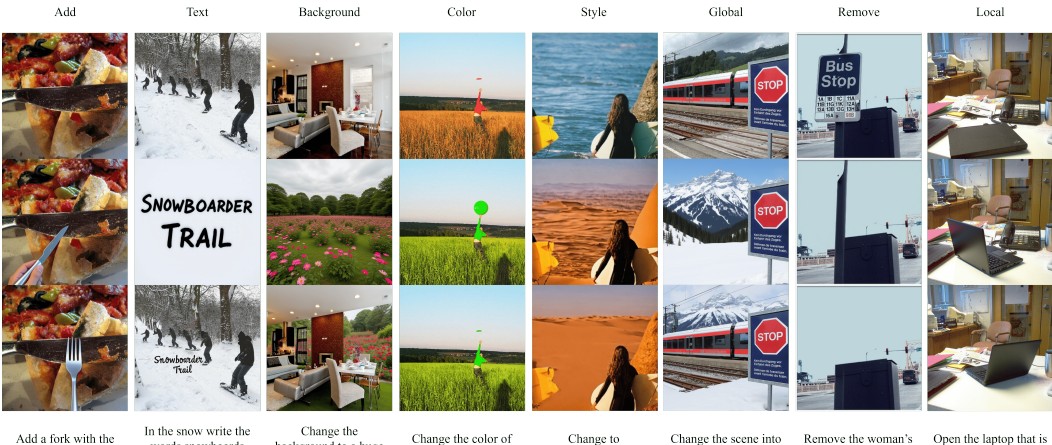

Figure D4: Visualization on Kontext comparing CFG and SWG. Rows from top to bottom: Source, CFG, and SWG.

## D.4 ALIGNEDEDIT ALGORITHM DETAILS

To provide a clear view of our approach, we present the algorithmic formulation of ALIGNEDEDIT. Our method is organized into two complementary components.

**Algorithm in a Single Transformer Block** We construct a semantic weak embedding as follows. Given image tokens and text embeddings, we first compute cross-attention to obtain the attention matrix. We apply a softmax over the text dimension to normalize the scores, then compute per-token saliency by averaging each token's attention across all image tokens and attention heads. We rank these saliencies and select the top-K tokens (the largest values). For the selected tokens, we attenuate their embeddings by a scalar s in (0,1) while leaving the others at 1, producing an attenuated text embedding. Using this attenuated text embedding, we recompute keys and values while keeping queries unchanged.

**Algorithm for Noise Prediction** The second algorithm uses the attenuated embedding obtained from Algorithm 1 to perform CFG-style noise prediction for image editing. At each diffusion step, the denoiser is evaluated three times: first, with the source image condition and the positive embedding (strong branch); second, with the source image condition and the attenuated embedding produced by applying Algorithm 1 to all transformer blocks (weak branch); and third, with the source image condition removed while still using the attenuated embedding (weak–null branch). Weighted fusion proceeds as follows. Take the third output as the baseline, then form two contrast signals: the image contrast, defined by the difference between the second and third outputs, which reinforces source structure; and the semantic contrast, defined by the difference between the first and second outputs, which drives the intended edit. The final prediction is obtained by adding to the baseline the image contrast scaled by an image-guidance weight and the semantic contrast scaled by a semantic-guidance weight. The image-guidance weight controls how much of the original layout and texture are preserved, while the semantic-guidance weight controls the strength of the edit. Using this combined prediction, perform one denoising step and iterate across the diffusion schedule to produce the final edited image.

---

**Algorithm 1: Single-Block Text Attenuation**

**Input** : scaling factor $s \in (0,1)$; Top-$K$ value $k$;
initial text embedding $E_0^T$ and text length $L_T$;
initial image embedding $E_0^I$;
initial latent $X_0$ and its length $L_x$;
Number of Attention Blocks $B$.
**Output:** fully-updated latent $X_B$
**for** $b \leftarrow 1$ **to** $B$ **do**
  $Q \leftarrow \text{ToQ}(E_b^T, X_b, E_b^I)$
  $K \leftarrow \text{ToK}(E_b^T, X_b, E_b^I)$
  $A_{\text{org}} \leftarrow \text{AttMap}(Q, K)$
  $A_{TL} \leftarrow A_{\text{org}}\left[\,0\!:\!L_T,\ L_T\!:\!L_T + L_X\,\right]$
  $\{i_1, \ldots, i_K\} \leftarrow$
    $\text{TopKIndices}\big(\max\_\text{over\_rows}(A_{TL}),\, k\big)$
  $E_b^{T'} \leftarrow \text{DeepCopy}(E_T)$
  **foreach** $j \in \{i_1, \ldots, i_K\}$ **do**
    $E_b^{T'}[j] \leftarrow s \cdot E_T[j]$
  $Q_{\text{new}} \leftarrow \text{ToK}(E_b^{T'}, X_b, E_b^I)$
  $K_{\text{new}} \leftarrow \text{ToK}(E_b^{T'}, X_b, E_b^I)$
  $V_{\text{new}} \leftarrow \text{ToV}(E_b^{T'}, X_b, E_b^I)$
  $A_{\text{weak}} \leftarrow \text{AttMap}(Q_{\text{new}}, K_{\text{new}})$
  $X_{b+1} \leftarrow X_b + A_{\text{weak}} V_{\text{new}}$
  **IF** Model *is* Kontext **THEN**
    $E_{b+1}^T \leftarrow E_b^{T'} + A_{\text{weak}} V_{\text{new}}$
**return** $X_B$

---

**Algorithm 2: SWG Sampling**

**Input** : initial latent $x_T$; timesteps $\mathcal{T}$;
    predictor $\epsilon_\theta$; image condition $c_I$;
text $c_T$; weak text $c_{\text{weak}}$; text guidance $g$; image guidance $g_I$.
**Output:** edited latent $x_0$
**for** $t \in \mathcal{T}$ **do**
  $S_{\text{null}} \leftarrow \epsilon_\theta(x_t, t, \varnothing, c_{\text{weak}})$
  $S_{\text{text}} \leftarrow \epsilon_\theta(x_t, t, c_I, c_T)$
  $S_{\text{img}} \leftarrow \epsilon_\theta(x_t, t, c_I, c_{\text{weak}})$
  $\hat{\epsilon}_t \leftarrow S_{\text{null}} + g\,(S_{\text{text}} - S_{\text{img}}) + g_I\,(S_{\text{img}} - S_{\text{null}})$
  $x_{t-1} \leftarrow \text{Step}(x_t, t, \hat{\epsilon}_t)$

Figure D5: Procedures of **ALIGNEDEDIT**: (1) token attenuation in a single transformer block (2) SWG sampling.

# E EVALUATION DETAILS

## E.1 USER STUDY DETAILS

We conducted a controlled user study with 30 participants to assess preferences for edited images. Each trial displayed one source image, one editing instruction, and three edited candidates (A/B/C) from different methods; candidates A, B, and C corresponded to a CFG-based baseline, a PostEdit-based method, and our method (ALIGNEDEDIT), respectively. Participants answered three single-choice questions about instruction fidelity, source preservation, and whether the edited image appears natural. Using a within-subjects design with 12 tasks per participant and 3 questions per task, we obtained 1,080 total responses. Task order and A/B/C model assignments were randomized and blinded to avoid presentation bias. All stimuli were sampled at random from Emu Edit and ImgEdit Bench. An example question from our user study is shown in Figure E1.

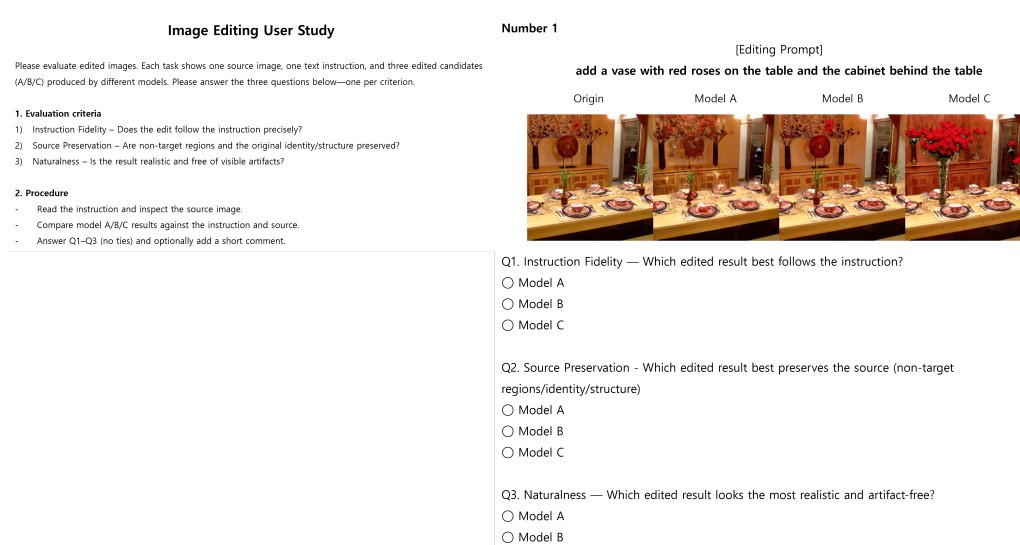

Figure E1: User study interface (example). Each task presents one source image and instruction (top), along with three edited candidates (A/B/C). Participants answer three single-choice questions per task: Q1 Instruction Fidelity, Q2 Source Preservation, Q3 Naturalness. Model labels (A/B/C) are randomized and blinded.

### E.2 MULTI-TURN EDITING PROTOCOL

Because our model operates in a single-turn setting, we slightly adapted the ImgEdit Bench benchmark to evaluate multi-turn editing. An example prompt is shown in Figure E2.

```
#    Curated Multiturn Image Editing Prompts

 1 content memory
    [T1]   Ensure all subsequent edits are yellow. Change the license plate color
    [T2]   Change the car body color to yellow
 2 content memory
    [T1]   Ensure all subsequent edits use leather material. modify the pillow's material
    [T2]   Add a piece of clothing on the bed
 3 content memory
    [T1]   Include coffee-related elements in all subsequent edits. add a cup
    [T2]   Randomly place some items next to the cup
 4 content memory
    [T1]   Ensure all subsequent edits use iron material. Change the door's material
    [T2]   Add a table in the open area
 5 content memory
    [T1]   Ensure all subsequent edits are milky white. Place a sculpture in the open area
    [T2]   Change the color of the flower bed
...
11 content understand
    [T1]   Change the license plate to black
    [T2]   Turn it white and write 111 on it
12 content understand
    [T1]   Add a ring of decorative garnishes to the plate
    [T2]   Turn ring of decorative garnishes black
13 content understand
    [T1]   Turn the pillow white
    [T2]   Turn pillow black
14 content understand
    [T1]   Replace the liquid in the teacup with milk
    [T2]   Change milk to clear water
15 content understand
    [T1]   Change the material of the door to iron
    [T2]   Turn material black
16 content understand
    [T1]   Remove the flower bed on the left
    [T2]   Add a trash can in flower bed
17 content understand
    [T1]   Change the sign to black lettering
    [T2]   Add the word 'hello'
...
21 backtrace version
    [T1]   Change the characters on the license plate to 1
    [T2]   Change the car body to white
22 backtr: Delete the tomato
    [T1]   Your edit is not good. redo it
    [T2]   Turn the quilt white
23 backtr: Add a piece of clothing on the bed
    [T1]   Add a cup
    [T2]   Randomly place some items next to the cup
```

Figure E2: Example prompt used in the multi-turn editing protocol.

## E.3 QUALITATIVE RESULTS

We additionally present qualitative results of ALIGNEDEDIT in Figure E3 to Figure E6. For a fair comparison, we juxtapose against CosXL and Flux without applying **SWG**.

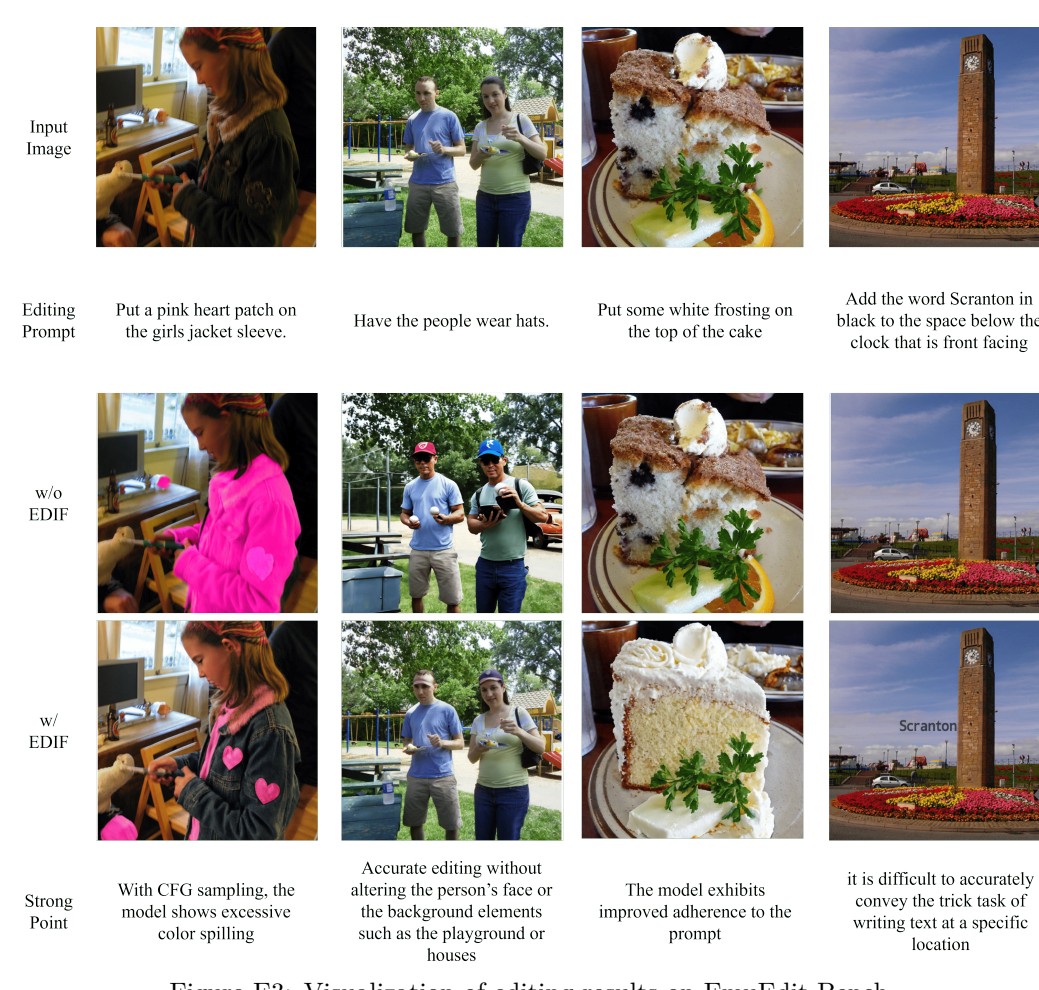

Figure E3: Visualization of editing results on EmuEdit-Bench.

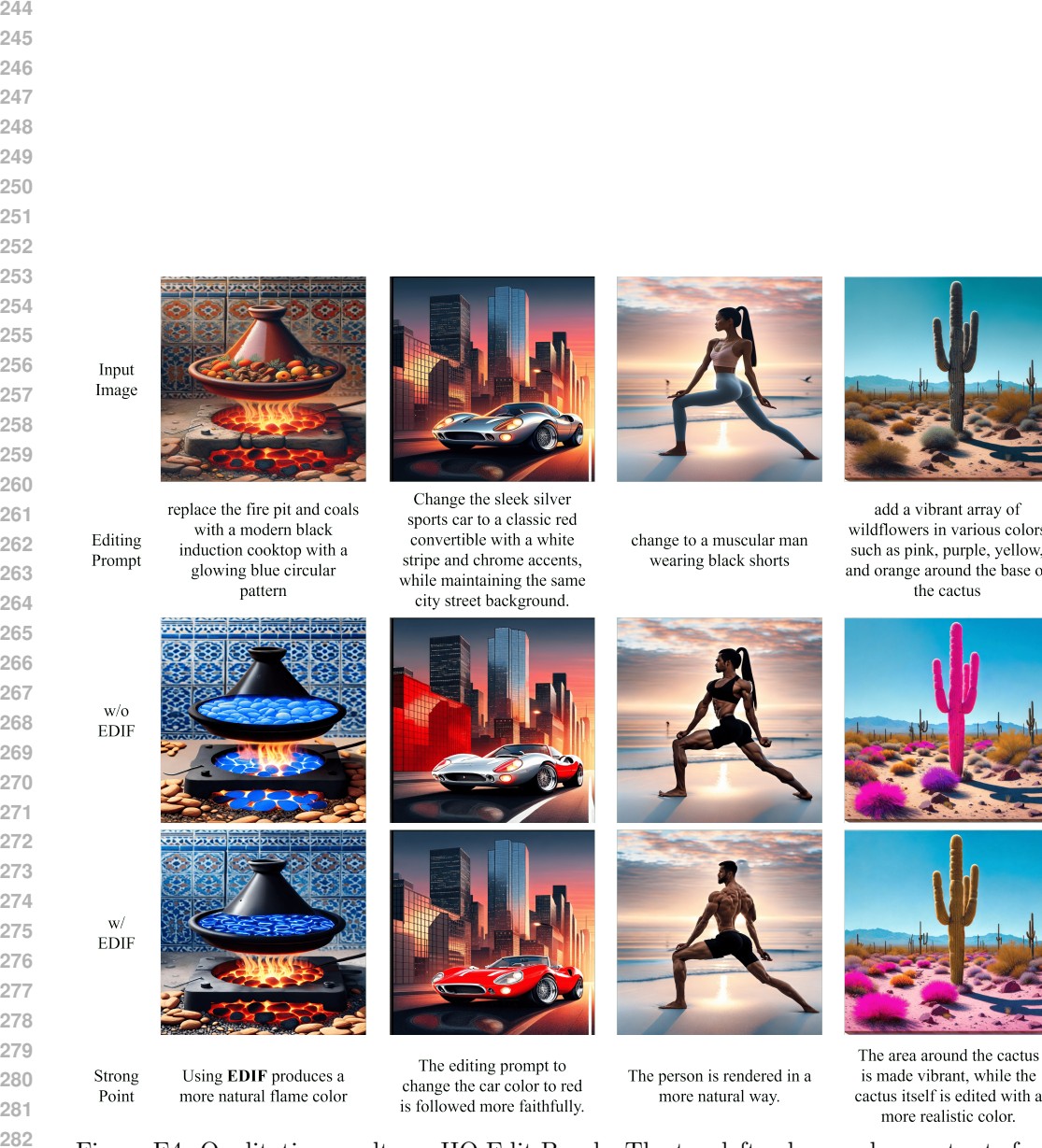

Figure E4: Qualitative results on HQ Edit Bench. The two left columns show outputs from CosXL (left with CFG sampling and the right with SWG sampling), and the two right columns show outputs from Kontext.

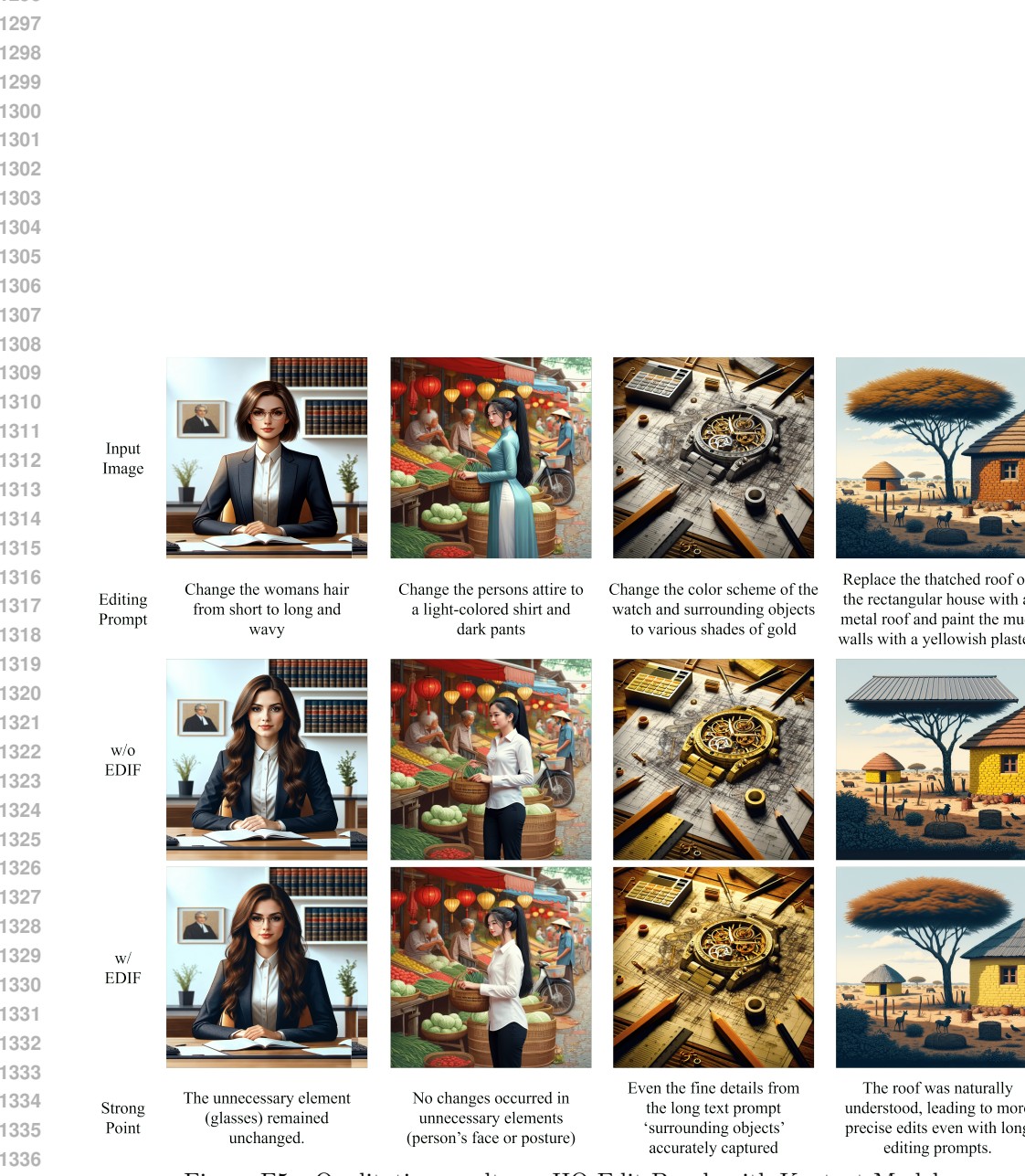

Figure E5: Qualitative results on HQ Edit Bench with Kontext Model.

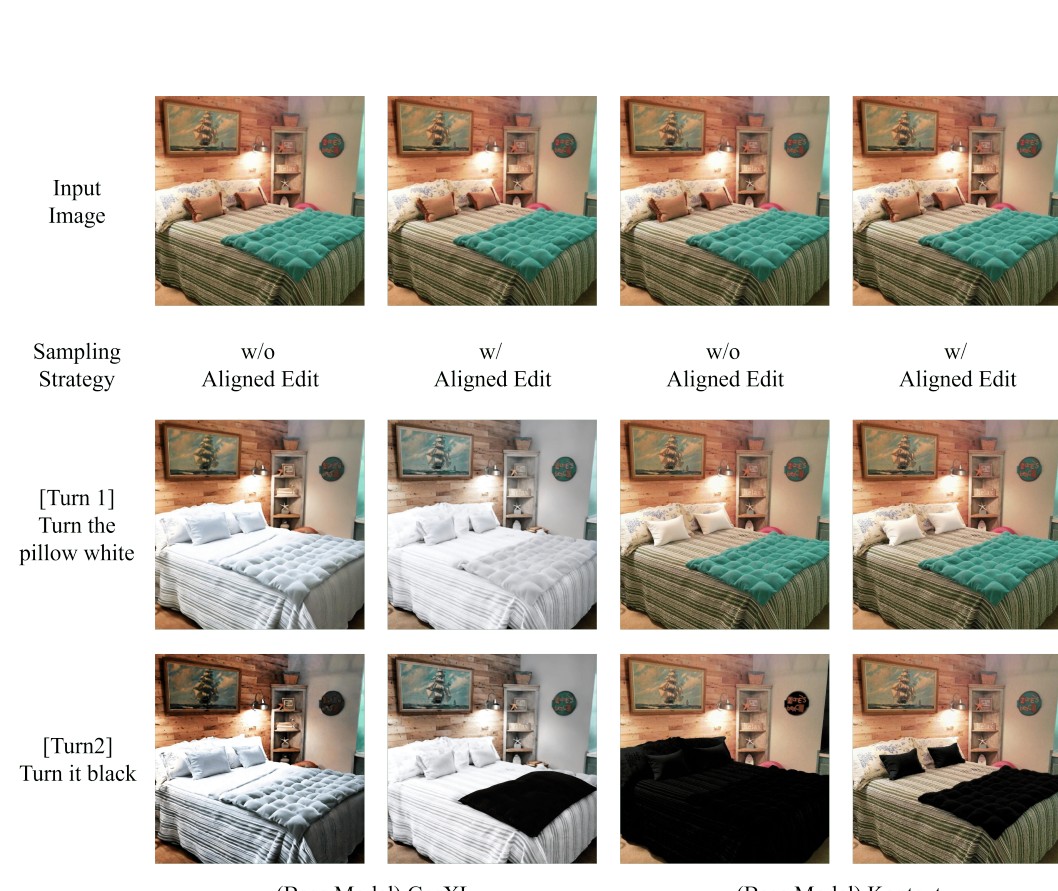

Figure E6: Visualization of multi-turn editing results on ImgEdit-Bench. The first two columns show CosXL with CFG sampling (left) and SWG sampling (right). In the left case, the model ignored the instruction 'change it to black', modifying the wall hanging instead and misinterpreting 'black' as a global darkening of the image. Columns three and four present results from the base model, Kontext, with CFG and SWG sampling, respectively. In the third column, the model again ignored the prompt and unnecessarily darkened additional elements, including the wall hanging.