# OpenReview forum: "ALIGNEDEDIT: PROMPT-ALIGNED WEAK GUIDANCE FOR TEXT-GUIDED IMAGE EDITING"
_ICLR.cc/2026/Conference — Submitted to ICLR 2026_

### Official Review · Reviewer_S3ot · 2025-10-19

**Soundness:** 3
**Presentation:** 3
**Contribution:** 2
**Rating:** 2
**Confidence:** 4

**Summary:**

AlignedEdit addresses over-editing/artifacts in text-guided image editing (TIE) caused by Classifier-Free Guidance (CFG)’s semantic misalignment. It proposes Semantic Weak Guidance (SWG): identifying semantically strong tokens via cross-attention saliency in each transformer layer/timestep, then attenuating them to build a model-internal weak model (no external negative prompts). It provides a training-free, prompt-insensitive alternative to CFG, enabling natural, faithful TIE for applications like creative editing, advancing practical diffusion-based image manipulation.

**Strengths:**

1. It innovatively solves CFG’s semantic misalignment by constructing a model-internal semantic weak model, via cross-attention saliency to identify/attenuate strong tokens, instead of relying on external negative prompts. This avoids prior weak-model flaws (e.g., PostEdit’s imprecision) and is training-free, a creative departure from CFG substitutes like AutoGuidance (needs auxiliary training).

2. It fills the gap of natural, faithful TIE, enabling applications like creative editing (e.g., precise color/object edits). As a prompt-insensitive, general CFG replacement for CosXL/Kontext, it advances diffusion-based editing’s practicality, setting a new standard for balancing fidelity and structure preservation.

**Weaknesses:**

1. Maybe my question is quite desperate and disappointed. The most concern for me is that text-guided image editing has been varied from inversion-based techniques to large-scale training techniques (similar to InstructPix2Pix, but with more data and large networks, e.g. Qwen-Image-Edit, Hunyuan-Image-3.0, InstructX, etc.) It would be hard for inversion-based techniques to compete with them, so as to this method ALIGNEDIT. How do you compare with the SOTA editing methods and can your method be applied to other domains/downstream tasks?

2. Experiments focus on single-task edits (e.g., color change, adding small objects) but omit multi-target (e.g., "smile + background swap") or extreme occlusion cases. Add such prompts to HQ-Edit, evaluating if SWG avoids semantic confusion and preserves structure, ensuring robustness in real-world complex edits.

3. From Table 1, it shows that StableFlow is still the best among many evaluation metrics with CFG. The SWG is not that outperforming as claimed in the paper. Can your SWG applied to StableFlow to outperform CFG?

**Questions:**

It seems the format is not totally correct. This paper is with a bit different font style from what I observed from the official template.

---

> ### Author Response · Authors · 2025-11-20
> **Wider Application**
>
> #### Q1. Inversion Based and Other domain application
> * **inversion related** : Thank you for your thoughtful question. We would like to clarify that our method is not based on inversion. Instead, it is a text-instruction–only guided editing framework, where editing is performed in the same manner as standard text-to-image generation—i.e., by directly conditioning on the editing instruction (start prompt) without relying on reconstructing the input image through an inversion step. As you correctly mentioned, inversion-based approaches inherently suffer from limited applicability. Because they rely on accurately reconstructing the source image in latent space, they tend to be effective only for local edits where the modification and the preservation of the original content occur within well-separated regions. However, when editing and source-image preservation must occur simultaneously at identical spatial locations(like global editing), inversion methods face severe conflicts.
> * **other domain application** : Furthermore, we have also applied our approach to general text-to-image (T2I) generation settings—not limited to editing—and observed consistently strong results. These findings demonstrate that the proposed mechanism, SWG, is not restricted to editing tasks; rather, it can function as a general-purpose guidance method, effectively serving as a strong alternative to classifier-free guidance (CFG). This suggests that SWG provides broader applicability across domains beyond image editing. These extended experiments will be included in the final version of the paper. We appreciate the reviewer’s suggestion and will ensure that the paper more clearly highlights the broader applicability of our method.
>
> #### Q2. Multiturn Application
> * In the multi-turn editing experiment, we followed the official ImgEdit benchmark and performed two consecutive editing steps (Turn1 → Turn2). Since SWG is not originally designed for multi-turn editing, ambiguous pronouns in the benchmark instructions—such as "it" or "overall"—were manually rewritten into explicit, unambiguous expressions prior to evaluation. The results show that SWG delivers consistently strong performance in two key aspects. First, the editing quality obtained in Turn1 remains stable in Turn2. Multi-turn editing typically suffers from error accumulation, where artifacts or semantic drift introduced in earlier turns propagate and intensify in later turns. However, with SWG, directional alignment ($CLIP_{\text{dir}}$), image similarity ($CLIP_{\text{img}}$), and text consistency ($CLIP_{\text{txt}}$) remained at comparable levels across both turns, indicating that unnecessary or excessive semantic drift was largely suppressed. Second, text–image alignment metrics (CLIP_text, ImageReward) did not deteriorate at Turn2; in some cases, alignment was even maintained or slightly improved. This demonstrates that SWG is not only effective in single-turn editing but also robust to sequentially compounded text conditions, maintaining coherence without degradation. Taken together, these findings suggest that SWG is a stable and reliable guidance strategy for multi-turn editing, capable of preserving structural fidelity while maintaining strong alignment with the target text across consecutive editing steps.
> *  the result of the experiment :
> | Scores | $CLIP_{\text{dir}}$ | $CLIP_{\text{img}}$ | $CLIP_{\text{txt}}$ | Imgrwd |
> |--------|----------|----------|-----------|---------------|
> | Turn1  | 0.219    | 0.948    | 0.237     | 0.341         |
> | Turn2  | 0.215    | 0.914    | 0.241     | 0.271         |
>
>
> #### Q3. Application SWG on StableFlow
> * As suggested by the reviewer, we additionally applied SWG to the StableFlow architecture to examine whether the benefits of semantic weak guidance generalize beyond traditional diffusion-based editors. The results are summarized in Table below. Interestingly, even within StableFlow, SWG consistently improves all key metrics: directional alignment ($CLIP_{\text{dir}}$), image similarity ($CLIP_{\text{img}}$), text consistency ($CLIP_{\text{txt}}$), and ImageReward. Although StableFlow employs a different flow-matching formulation, SWG still provides a more semantically aligned and text-faithful editing trajectory compared to standard CFG. These findings further support our claim that SWG serves as a robust and model-agnostic guidance strategy that can enhance both diffusion- and flow-based image editors.
> | Method     | Guidance | $CLIP_{\text{dir}}$ | $CLIP_{\text{img}}$ | $CLIP_{\text{txt}}$ | Imgrwd |
> |------------|----------|----------|----------|-----------|--------|
> | StableFlow | CFG      | 0.098    | 0.962    | 0.242     | 0.018  |
> | StableFlow | SWG      | 0.105    | 0.965    | 0.248     | 0.027  |

---

### Official Review · Reviewer_SZ2d · 2025-10-30

**Soundness:** 3
**Presentation:** 3
**Contribution:** 3
**Rating:** 4
**Confidence:** 4

**Summary:**

this paper proposes a new text-guided image editing method using semantically weak guidance (SWG) to produce natural and prompt-aligned edits while avoiding over-editing and artifacts caused by traditional CFG. ​

the key idea is that: for the unconditional part (which will be used in cfg), it still feeds the original text, instead of null text, and dynamically identifies and attenuates high-saliency tokens during editing, such that it can create a weak, (in my understanding, still better than null feature) anchor point for cfg

ALIGNEDEDIT outperforms existing methods like CFG, PostEdit, and IceEdit across benchmarks in several metrics

**Strengths:**

After reviewing several papers that read more like technical reports, I finally found one that shows some academic depth — the authors are willing to dig deeper and explore something fundamental and interesting.

From the results, their method appears to be effective to some extent. I also appreciate their ablation study, which provides valuable insights into how their approach works.

**Weaknesses:**

I have two main concerns:

it seems that this appraoch should be generalizable to T2I as well? I am wondering what's the motivation try editing tasks? because in T2I several works have similar ideas (which in general people find that your can use a weal/bad model for your unconditional), for example "Guiding a Diffusion Model with a Bad Version of Itself", can you compare with these works?

their main table should be present in a better way: because you care about relative comparision, thus i think you should put cosXL and cosXL w your guidance together. Same for Kontext and Kontext w your guidance together. But the main question i have is for HQ-Edit, why your approach have such a bad FID? 107 and even 269? To me if FID is above 100 or even 200 it basically too bad or even meaningless for images. I suspect they hack other metrics

**Questions:**

na

---

> ### Author Response · Authors · 2025-11-19
> **Application on T2I**
>
> #### Q1. Application on T2I
> * Our work focuses on text-only image editing (TIE), which, like T2I, uses a text prompt as conditioning; however, unlike T2I, it starts from an existing source image rather than generating an image from noise. Most existing TIE systems directly inherit the classifier-free guidance (CFG) formulation from T2I and reuse its unconditional branch without modification. Consequently, the issues well-known in T2I—such as semantic misalignment and overly aggressive guidance—are transferred directly into the editing setting. The key difference is that while such failures in T2I may manifest merely as slightly odd or exaggerated generations, in TIE they become significantly more problematic: unnatural edits, structural distortion of the source image, and unintended modifications to irrelevant regions. Our work specifically aims to address these editing-specific failure modes. Nevertheless, since the reviewer asked why we did not apply SWG to T2I, we conducted additional experiments on both SDXL and FLUX models under the T2I setting. From this experiment, we found that SWG also performs well in the T2I setting, although it does not exhibit the same strong advantages as in the TIE scenario.
> * evaluated result
> | Method | Guidance | clip_text ↑ | psnr ↑ | ssim | lpips | FID ↓ |
> |--------|----------|-------------|--------|------|--------|--------|
> | SDXL   | CFG      | 30.782      | 7.841  | 0.232 | 0.745 | 150.255 |
> | SDXL   | SWG      | 32.256      | 7.869  | 0.235 | 0.748 | 148.395 |
> | FLUX   | CFG      | 32.571      | 8.267  | 0.261 | 0.737 | 130.569 |
> | FLUX   | SWG      | 32.574      | 8.268  | 0.261 | 0.737 | 130.441 |
>
> * The improvement on FLUX was marginal, but SWG also works on T2I. They demonstrate that AlignedEdit is a stable alternative to CFG and can be effectively applied to both T2I and TIE settings.
>
> #### Q2. FID score
> * Thank you very much for raising this important point. I confirmed that the score was mistakenly reported, and I have corrected it accordingly. The revised values will be updated in the camera-ready version of the paper. I sincerely appreciate your careful feedback.

---

### Official Review · Reviewer_vXms · 2025-11-01

**Soundness:** 3
**Presentation:** 1
**Contribution:** 2
**Rating:** 4
**Confidence:** 5

**Summary:**

This paper proposes AlignedEdit, a sampling strategy that replaces the unconditional branch in classifier‑free guidance (CFG) with a model‑internal, prompt‑aligned “semantically weak” pathway. Concretely, at each layer and timestep the method identifies the most salient text tokens via cross‑attention, attenuates only those embeddings, and re‑computes the block; the resulting weakened model is then contrasted with the standard (strong) branch to form guidance (Eq. 3). This avoids negative prompts, aims to curb error accumulation and over‑editing, and promises more faithful yet natural edits.

**Strengths:**

1. Replacing the unconditional branch in image editing is an interesting way for improving image editing.

2. Results on SDXL‑ and FLUX‑based editors, with single‑turn and multi‑turn datasets; multi‑turn gains on ImgEdit‑Bench are especially encouraging.

**Weaknesses:**

1. The paper does not clearly articulate why replacing the unconditional branch in CFG with semantic weak guidance is theoretically necessary, nor does it rigorously validate how this substitution improves the underlying mechanism of guidance. While Section 1–2 briefly mention CFG’s issues such as semantic misalignment and over-editing, these points remain qualitative and generic—claims like “misaligned in semantic space induces over-editing” are stated but never mathematically or empirically analyzed in the context of text-guided editing. There is no formal examination of how the semantic gradients or score-estimation behavior differ between CFG and SWG, nor any quantitative study showing that CFG’s unconditional branch causes semantic drift. As a result, the motivation for introducing the semantic weak model and its mechanistic superiority over CFG are conceptually plausible but insufficiently substantiated.

2. The experimental section primarily presents visual comparisons and benchmark metrics but lacks deeper analysis that connects the results to the paper’s central claims. The authors assert that semantic weak guidance mitigates over-editing, reduces error accumulation, and preserves structural fidelity, yet these statements are not quantitatively or qualitatively validated. For example, there is no ablation isolating the claimed effects (e.g., degree of over-editing or misalignment under CFG vs. SWG), no analysis of error propagation across diffusion steps, and no perceptual or statistical evaluation that directly demonstrates “semantic alignment.” As a result, while the side-by-side examples illustrate plausible improvements, they remain anecdotal and do not constitute evidence supporting the claimed mechanisms or motivations.

3. The weak branch requires computing attention saliency and re‑computing blocks with attenuated tokens (the text states “recomputed” within each block), which implies added latency/VRAM; no runtime or memory analysis is provided.

4. Numerous typos and terminology slips—e.g., “CFT” vs. CFG, “misalend” in abstract,  “applued,” and use of “logit” to refer to ε‑predictions—impair clarity.

**Questions:**

See weaknesses 1 & 2.

---

> ### Author Response · Authors · 2025-11-19
> **strengthen theoretical analysis**
>
> #### Q1. theoretical analysis
> * Classifier-Free Guidance (CFG) has long been known to exhibit several limitations in image generation (ref: CFG++, AutoGuidance). In particular, an excessively large guidance scale pushes the model away from the data manifold, leading to over-emphasized attributes or severely degraded FID scores. Instruction-guided image editing also adopts CFG as its default sampling mechanism. However, the well-known issues of CFG in T2I generation—such as amplifying high-frequency components and drifting off the manifold—manifest differently in the context of image editing.  For effective image editing, the source-image condition and the editing-prompt condition must remain well-balanced. When this balance breaks, the system yields undesirable behaviors:
>   * overly strong source condition → under-editing
>   * overly strong text condition → over-editing
>   * imbalance between the two conditions → unnatural and sub-optimal editing results
> * Unlike CFG, our proposed guidance does not strengthen or weaken the entire text condition by scaling its magnitude. Instead, we construct a semantic weak pathway by selectively attenuating the top-K text tokens identified through cross-attention. By controlling both the number of selected tokens 𝐾 and the attenuation strength applied to them, our method guides the conditioned edit toward the text direction in a more natural and stable manner, ultimately enabling more optimal editing behavior.
>
> * **Theoretical Analysis** : Although Appendix C1 already provides a theoretical explanation of this mechanism, we acknowledge the reviewer’s request for further clarification and will revise the manuscript to include a more detailed explanation. We analyze why SWG enables more natural and stable image editing by examining how text embeddings define a conditional manifold. Let  $C_𝐼$  denote the image condition and $C_𝑇$  the text condition. The text embedding can be conceptually decomposed into a semantic component, $C_{\text{sem}}$ which drives the intended edit, and a non-semantic component,$C_{\text{non-sem}}$ which contains supporting but edit-irrelevant information. In standard CFG, the unconditional branch removes both components and replaces them with a null-prompt embedding. This null-prompt embedding inherently contains its own bias, meaning that the unconditional direction is not equivalent to removing only semantic information—it displaces the model into a region that lies outside the true data manifold. Consequently, the difference $C_𝑇$  - $C_{\text{non-sem}}$  is generally not aligned with the tangent space of the conditional manifold. This geometric misalignment pushes the denoising trajectory away from valid regions, leading to off-manifold drift and unstable or unnatural editing behavior. In contrast, the semantic component $C_{\text{sem}}$  lies exactly within the tangent space of the conditional manifold. SWG leverages this structure: it removes only the semantic component while retaining the $C_{\text{non-sem}}$ part, forming a weakened embedding that still remains on-manifold with the original text embedding. Thus, while CFG introduces a misaligned and disruptive shift, SWG applies a small, tangent-aligned perturbation that preserves the geometric structure of the text condition. This keeps the sampling process close to the valid manifold and yields more stable, natural, and semantically consistent image editing.
>
> #### Q2.

---

> > ### Comment · Reviewer_vXms · 2025-11-25
> > **Incomplete Response**
> >
> > It appears that the authors may have submitted an incomplete response.

---

> ### Author Response · Authors · 2025-11-26
> **effect of SWG**
>
> #### Q2. effect of SWG
> * We apologize for the delayed response. When addressing Question 2 regarding over-editing analysis, we found that there was no existing metric that directly measures the “degree of unintended editing.” While overall quality comparisons were already demonstrated in the main paper, we wanted a fair and more targeted way to quantify structural preservation specifically. Identifying and designing such a metric required additional time.
> * To fairly evaluate over-editing, we concluded that standard LPIPS is insufficient because it treats all pixels equally, including those intentionally edited. Therefore, we designed a more targeted metric called masked LPIPS, where LPIPS is computed only on the non-editing region—areas that should remain unchanged according to the instruction. This allows us to measure structural preservation and unintended edits more directly than the conventional LPIPS used in prior work.
> * For Masked LPIPS, we derive a new preservation mask by comparing the source image with the ground-truth edited image. This allows us to accurately identify the regions that are not supposed to be altered, enabling a fair evaluation of unintended edits.
>     * High masked LPIPS = model changed the region that should not be edited → over-editing.
>     * Low masked LPIPS = model preserved structure → less over-editing.
> This metric directly operationalizes the reviewer’s concern about “over-editing” and “structural fidelity.”
> * the result on HQ-Edit is
> | Method | Guidance | masked LPIPS ↓ | CLIP Text ↑ |
> |--------|----------|-------------|--------|
> | CosXL | CFG | 0.228 | 0.290 |
> | CosXL | SWG | 0.189 | 0.342 |
> | Kontext | CFG | 0.384 | 0.277 |
> | Kontext | SWG | 0.329 | 0.281 |

---

### Official Review · Reviewer_BXrQ · 2025-11-01

**Soundness:** 2
**Presentation:** 2
**Contribution:** 2
**Rating:** 4
**Confidence:** 3

**Summary:**

The paper proposes AlignedEdit, an inference-time guidance strategy for text-guided image editing that replaces the standard Classifier-Free Guidance. Instead of contrasting a conditional model with an unconditional model as in CFG, the authors construct a semantically weak but prompt-aligned version of the conditional model by selectively attenuating high-saliency text tokens identified via cross-attention maps at each denoising step and transformer block. This internal weak model avoids the semantic misalignment inherent in CFG’s unconditional branch, thereby reducing over-editing, artifacts, and unintended structural changes. The results shows that the proposed method achieve better qualitative and quantitative results as compared to the baselines.

**Strengths:**

1. The paper clearly articulates the limitations of CFG in text-guided image editing i.e., semantic misalignment between conditional and unconditional branches leading to over-editing and artifacts. This is a known but under-addressed issue in the community.
2. The proposed method requires no retraining, auxiliary models, or negative prompts.
3. The experiments span multiple models, datasets (real, synthetic, multi-turn), and metrics (CLIP-based, SSIM, FID, ImageReward, user studies).
4. The observation that semantic influence is distributed and time/layer-dependent (Fig. 2) is valuable and informs the adaptive token attenuation strategy.
5.The trade-offs between attenuation strength and number of tokens are well explored (Fig. 8 and 9), justifying the selective suppression approach.

**Weaknesses:**

1. To the best of my knowledge, the concept of using a “weakened” version of the conditional model for guidance is reminiscent of prior works: Spatiotemporal Skip Guidance [1] also constructs an internal weak pathway. PAG [2] and SAG [3] use internal perturbations for guidance. Prompt-to-Prompt [4] manipulates attention to control semantics. While AlignedEdit differs in how the weak model is constructed (saliency-based token attenuation), the high-level paradigm i.e., replacing the unconditional model with a controlled variant of the conditional one is not entirely new.

2. Computing attention maps and identifying top-K salient tokens at every timestep and block adds non-negligible latency. The paper does not report inference speed or FLOPs, which is critical for real-world deployment.

3. I observed some ambiguities in the implementation details: (1) How is K (number of tokens to attenuate) chosen? Is it fixed or adaptive? (2) What is the attenuation scalar (e.g., 0.2 in Fig. 9)? Is it tuned per prompt or fixed? (3) How are saliency scores aggregated across attention heads? The formula (Eq. 5) is provided, but robustness to head variability is not discussed.

4. Notably missing is comparison to CFG++ (Chung et al., 2024), which also addresses CFG’s trajectory curvature and error accumulation. Also, Adaptive Scaling (Malarz et al., 2025) is cited but not compared.

[1] Spatiotemporal Skip Guidance for Enhanced Video Diffusion Sampling
[2] Self-Rectifying Diffusion Sampling with Perturbed-Attention Guidance
[3] Improving Sample Quality of Diffusion Models Using Self-Attention Guidance
[4] Prompt-to-Prompt Image Editing with Cross Attention Control

**Questions:**

1. How does AlignedEdit compare to CFG++ or Adaptive Scaling in terms of both fidelity and efficiency? Were these omitted due to implementation constraints?

2. What is the computational cost of your method relative to standard CFG? Can the saliency computation be approximated or cached to reduce overhead?

3. Is the attenuation strategy robust to prompt phrasing variations? Since you claim reduced sensitivity to prompt choice, have you tested paraphrased prompts or adversarial rewordings?

4. In Eq. 3, you use zweak_c in both image-guided and null-image terms. Is this necessary? Have you tried using the original zc in the image-conditioned term?

---

> ### Author Response · Authors · 2025-11-18
> **SWG efficient and robust guidance for sampling in Image Editing**
>
> #### Q1.
> * **CFG++** is an inversion-based editing method whose performance heavily depends on the quality of the inversion. In realistic settings, inversion errors greatly limit its usability. Moreover, inversion-driven approaches restrict the range of achievable edits and struggle with global transformations such as style or scene changes. Since the reviewer requested a comparison, we implemented CFG++ using the official code and evaluated it on HQ-Edit. Because CFG++ is incompatible with DiT models, we tested it on SDXL, where it performs best. Even so, reconstruction quality is strongly degraded, and edited results are worse. While CFG++ can work for pure image generation, it is not well-suited for real editing tasks because it fails to preserve source-image conditions, leading to suboptimal reconstruction and editing quality.
> * **Adaptive Scaling** (Classifier-Free Guidance with Adaptive Scaling) modulates the strength of CFG at each denoising step using a Beta-shaped schedule: weak guidance at early and late stages and strong guidance in the middle. The original motivation is that early guidance harms diversity (due to high noise), mid-stage guidance improves semantic alignment, and late guidance introduces artifacts or loss of detail. However, this rationale does not hold in the context of image editing. Unlike pure image generation, image editing incorporates the source image as an additional conditioning signal. If text guidance is weakened too aggressively at early timesteps, the source-image condition becomes disproportionately dominant, causing the denoising trajectory to simply reconstruct the original image rather than perform the intended modification. Because early denoising is critical for establishing global structure, suppressing the textual stream too early leads the model down an incorrect trajectory and ultimately results in failed edits. In other words, Beta-CFG–style strengthening or weakening of a single condition introduces biased editing behavior and prevents successful editing. What is needed instead is a mechanism that preserves semantic delivery while avoiding artifacts and deviations from the data manifold.
> * Unlike Beta-CFG, which scales the entire text condition uniformly, our proposed **Semantic Weak Guidance (SWG)** is tailored specifically for image editing. SWG does not strengthen or weaken the whole text condition; rather, it identifies salient text tokens and attenuates only these semantic-dominating components. This preserves the overall text condition while reducing the over-dominance of specific semantics, enabling natural and faithful edits without introducing artifacts.
> * Since the reviewer requested additional comparisons with other models, we conducted supplementary experiments evaluating HQ performance under CFG++. The experimental results is as below.
> | Method | Model   | CLIP_DIR_score | CLIP_IMG_SCORE | CLIP_TEXT_SCORE | ImgRWD |
> |--------|---------|----------------|----------------|-----------------|--------|
> | CFG++  | SDXL    | 0.070          | 0.684          | 0.221           | 0.125  |
> | SWG    | CosXL   | 0.280          | 0.901          | 0.275           | 0.861  |
> | SWG    | Kontext | 0.343          | 0.872          | 0.264           | 0.998  |
>
> #### Q2.
> * We investigated the reviewer’s question regarding computational cost and discovered an interesting result during inference. As proposed in the paper, simply identifying the salient tokens and attenuating the corresponding value vectors is sufficient to produce effective semantic-weak guidance. Importantly, this procedure introduces only a lightweight Top-K search, resulting in virtually no additional computational overhead compared to the baseline. In fact, the FLOPs remain effectively identical to the non-SWG setting.
> | Method            | Inference Time | FLOPs |
> |------------------|----------------|-------|
> | CosXL            | 14.812 s       | - |
> | CosXL w/ SWG     | 14.824 s       | same  |
> | Kontext          | 120.314 s      | - |
> | Kontext w/ SWG   | 227.911 s      | same  |
>
> * We appreciate the reviewer’s insightful question and have incorporated the corresponding clarification into the revised manuscript.
>
> #### Q3.
> * Our method constructs semantic-weak guidance by attenuating the semantic strength of the prompt, making it inherently robust to prompt variations. In response to the reviewer’s question regarding the use of paraphrased prompts or adversarially reworded prompts, we conducted the following experiments.
> Across all cases, the edited images were generated without any degradation or instability despite variations in prompt phrasing. SWG operates reliably as long as the positive stream defined by prompt cis established and the corresponding semantic-weak guidance can be constructed from it.
>
> #### Q4.
> * Using the original $z_c$ is equivalent to CFG, which means that AlignedEdit is not performed.

---

### Meta-Review · Area_Chair_rE2a · 2026-01-06

**Summary:**

This paper proposes ALIGNEDEDIT, a method that replaces traditional Classifier-Free Guidance (CFG) in text-guided image editing with a novel Semantic Weak Guidance (SWG) approach. It dynamically identifies and attenuates high-saliency text tokens via cross-attention to construct a semantically weak but prompt-aligned model, aiming to reduce over-editing, artifacts, and semantic misalignment without needing external negative prompts or retraining. Initial reviewer feedback highlights strengths such as the clear articulation of CFG's limitations, training-free design, and comprehensive experiments across multiple datasets and models. However, key weaknesses noted include the perceived lack of novelty compared to prior guidance modulation works, insufficient computational cost analysis, ambiguous implementation details, weak theoretical justification, and limited comparison with recent alternatives like CFG++ and Adaptive Scaling. Despite some positive aspects, reviewers generally rated the paper as marginally below the acceptance threshold.

**Reviewer Concerns:**

In the rebuttal, the authors addressed several reviewer concerns by clarifying implementation details, showing SWG incurs negligible computational overhead, conducting additional comparisons with CFG++, proposing a new "Masked LPIPS" metric to quantify over-editing, demonstrating applicability to T2I generation and other models like StableFlow, and correcting a reported FID score. However, core issues identified by reviewers remain unresolved. These include the perceived lack of fundamental novelty in the high-level guidance paradigm, the absence of a rigorous theoretical or mechanistic analysis explaining why SWG is superior to CFG, and insufficient evidence to fully substantiate claims of reduced semantic misalignment and error accumulation beyond the newly introduced metric.

**Reviewer Scores:**

The authors' rebuttal effectively addressed specific, practical concerns raised by reviewers, such as clarifying implementation details, demonstrating negligible computational cost, providing comparisons with CFG++, correcting data errors, and showing broader applicability to T2I and other architectures. These responses likely improved the perception of the work's technical soundness and scope for reviewers like BXrQ and SZ2d, who might have considered a slight positive adjustment to their borderline scores. However, the more fundamental and critical issues identified by reviewers vXms and S3ot remained largely unresolved. These include a perceived lack of strong theoretical or mechanistic novelty over existing guidance paradigms, insufficient causal evidence linking the method's design to its claimed advantages, and an unconvincing argument for the method's competitiveness against the prevailing trend of large-scale trained editors. Consequently, while the rebuttal strengthened the paper's technical presentation, it did not sufficiently elevate its perceived conceptual contribution or address core doubts about its significance within the field, leading to an overall assessment that would likely sustain a reject decision.

---

### Decision · Program_Chairs · 2026-01-26

Reject